# Simultaneous macroscale and microscale wave–ion interaction in near-earth space plasmas

Z.-Y. Liu [1], Q.-G. Zong [1,2] ✉, R. Rankin [3], H. Zhang [4], Y. F. Wang [1], X.-Z. Zhou[1], S.-Y. Fu[1], C. Yue [1], X.-Y. Zhu[1], C. J. Pollock[5], S. A. Fuselier[6,7] & G. Le[8]

Identifying how energy transfer proceeds from macroscales down to microscales in collisionless plasmas is at the forefront of astrophysics and space physics. It provides information on the evolution of involved plasma systems and the generation of high-energy particles in the universe. Here we report two cross-scale energy-transfer events observed by NASA's Magnetospheric Multiscale spacecraft in Earth's magnetosphere. In these events, hot ions simultaneously undergo interactions with macroscale ($\sim 10^5$ km) ultra-low-frequency waves and microscale ($\sim 10^3$ km) electromagnetic-ion-cyclotron (EMIC) waves. The cross-scale interactions cause energy to directly transfer from macroscales to microscales, and finally dissipate at microscales via EMIC-wave-induced ion energization. The direct measurements of the energy transfer rate in the second event confirm the efficiency of this cross-scale transfer process, whose timescale is estimated to be roughly ten EMIC-wave periods about (1 min). Therefore, these observations experimentally demonstrate that simultaneous macroscale and microscale wave-ion interactions provide an efficient mechanism for cross-scale energy transfer and plasma energization in astrophysical and space plasmas.

Energy-transfer processes in hydrodynamics and magnetohydrodynamics are of fundamental importance[1,2] but are relatively poorly understood. In ordinary fluids and gases (e.g., the terrestrial atmosphere), thermal collisions mediate the conversion of macroscopic ordered energy to microscopic scales at which energy is dissipated[1]. However, most astrophysical and space plasmas are collisionless due to their very low density[2]. They experience cross-scale energy transfer through the action of long-range electromagnetic forces. The currently most accepted model of cross-scale energy transfer in collisionless plasmas is the turbulent local cascade model, which assumes energy transfer proceeds via a cascade across similar spatial scales[3,4].

Besides turbulent cascade, wave–particle interactions are also suggested to be able to mediate energy transfer processes in plasmas. However, there are numerous types of wave–particle interactions[2]. Their efficacy is mostly unknown. Until the beginning of NASA's Magnetospheric Multiscale (MMS) mission[5], the primary difficulty has been the paucity of measurements resolving the gyromotion, and the limited time-resolution of particle instruments onboard spacecraft. However, this situation has changed through the launch of MMS which comprises four identical spacecraft equipped with instruments capable of detecting temporal variations on ion-gyration scales.

Here, we consider another model of cross-scale energy transfer in collisionless plasmas—the cross-scale wave–particle interaction

[1]Institute of Space Physics and Applied Technology, Peking University, Beijing, China. [2]Key laboratory of solar activity and space weather, National Space Science Center, Chinese Academy of Sciences, Beijing, China. [3]Department of Physics, University of Alberta, Edmonton, AB, Canada. [4]Physics Department & Geophysical Institute, University of Alaska Fairbanks, Fairbanks, AK, USA. [5]Denali Scientific, Fairbanks, AK, USA. [6]Southwest Research Institute, San Antonio, TX, USA. [7]Department of Physics and Astronomy, University of Texas at San Antonio, San Antonio, TX, USA. [8]NASA Goddard Space Flight Center, Greenbelt, MD, USA. ✉e-mail: qgzong@pku.edu.cn

model. We show two events observed by MMS in Earth's magnetosphere. Analysis of the observed wave fields and ion velocity distribution functions reveals that hot ions in the two events simultaneously undergo interactions with macroscale (~$10^5$ km) ultra-low-frequency (ULF) waves[6] and microscale (~$10^3$ km) electromagnetic ion cyclotron (EMIC) waves[2]. As a result of the interactions, energy directly flows from macroscales down to microscales, or more precisely, from fluid scales to ion-gyration scales. The observations presented here confirm that cross-scale wave–particle interactions are an efficient mechanism for cross-scale energy transfer and collisionless plasma energization.

## Results
### The first event

We first investigate an event mediated by hydrogen ions ($H^+$), which are the dominant ion species of the Earth's magnetosphere in terms of number density. This event was observed in the duskside magnetosphere (GSE [4.1, 10.2, 0.0] Earth radius, L-shell=12.2 and magnetic local time = 18.0 h) on September 5, 2015. At this time, MMS was located 10.5° south of the magnetic equator (taken as the minimum-B point given in the MMS/Magnetic Ephemeris Coordinates data). In this event, the magnetosphere was quiet, with the Dst (Disturbance storm-time) index of about −11 nT and the AE (Auroral Electrojet) index less than 300 nT.

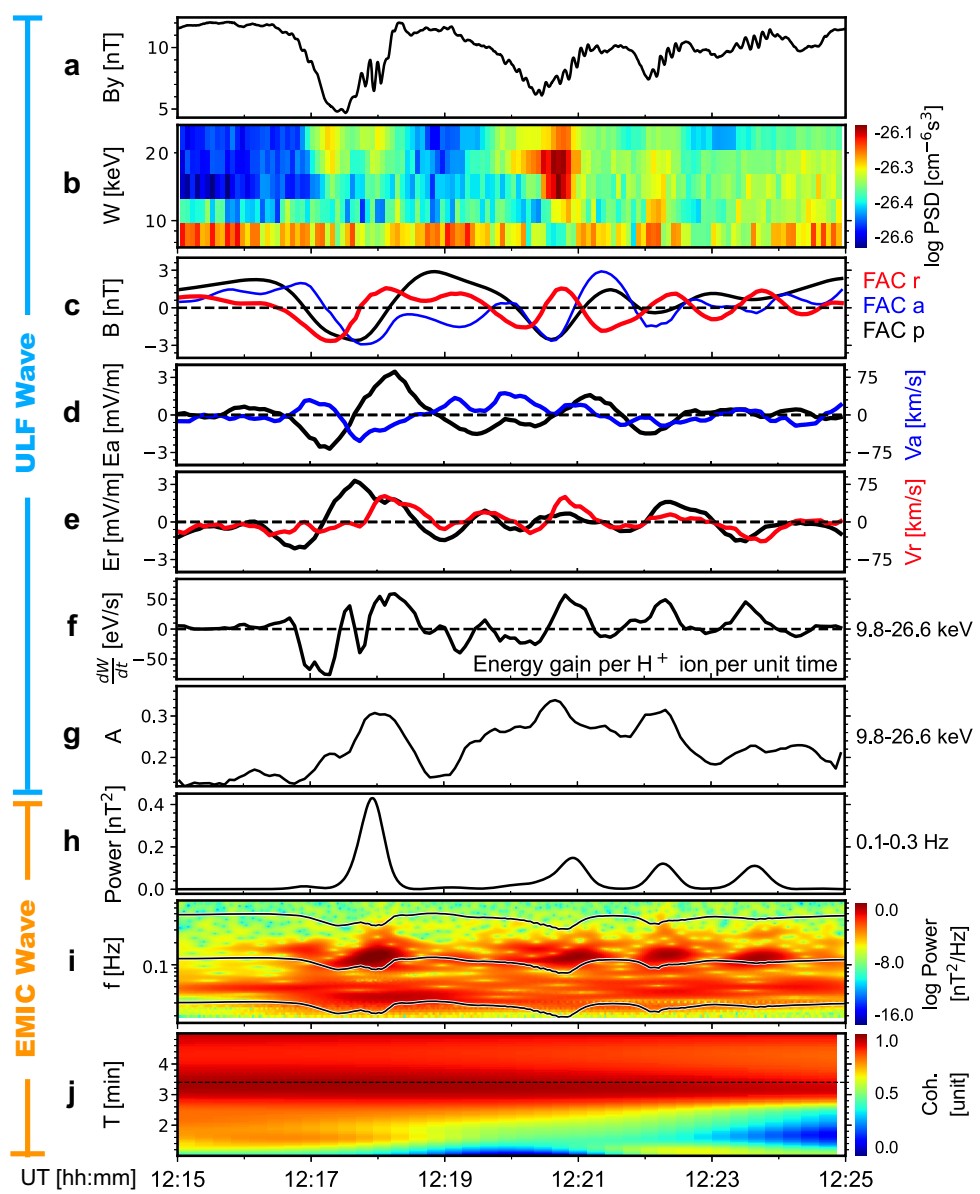

**Fig. 1 | Cross-scale interactions of ULF waves, EMIC waves and $H^+$ ions in the September 5, 2015, event.** Fast mode data is used here. **a** The GSE-Y component of the magnetic fields. **b** The energy-time spectrogram of the PSDs of 75°–105° PA ions from FPI. **c** ULF-wave magnetic field in the FAC system. The blue, green, and black curves correspond to the radial, azimuthal and parallel components, respectively. **d** The azimuthal component of the ULF-wave electric field (black) and the bulk velocity of 9.8–26.6 keV ions from FPI (blue). **e** The radial component of the ULF-wave electric field (black) and the bulk velocity of 9.8–26.6 keV ions from FPI (red). **f** $H^+$-ion energy gain from ULF waves per ion per unit time, averaged over the energy range of 9.8–26.6 keV. When generating panels (**c**–**f**), a 0.25–7 min bandpass filter has been used. **g** The anisotropy of 9.8–26.6 keV ions, defined as $(PSD_\perp - PSD_\parallel)/(PSD_\perp + PSD_\parallel)$, where $PSD_\parallel$ and $PSD_\perp$ represent the PSDs for parallel motion (0°-30° and 150°-180° PA) and perpendicular motion (75°-105° PA). **h** EMIC wave power integrated over 0.11–0.3 Hz. **i** The dynamic spectra of EMIC-wave magnetic field. From top to bottom, the three black curves represent the gyro-frequency of $H^+$, $He^+$, and $O^+$ ions. Power enhancements corresponding to EMIC waves can be seen between $H^+$ and $He^+$ gyro-frequency. **j** The cross-wavelet correlation coefficient between the ULF-wave field (panel **a**) and the EMIC-wave power integrated over 0.1–0.3 Hz (panel **h**).

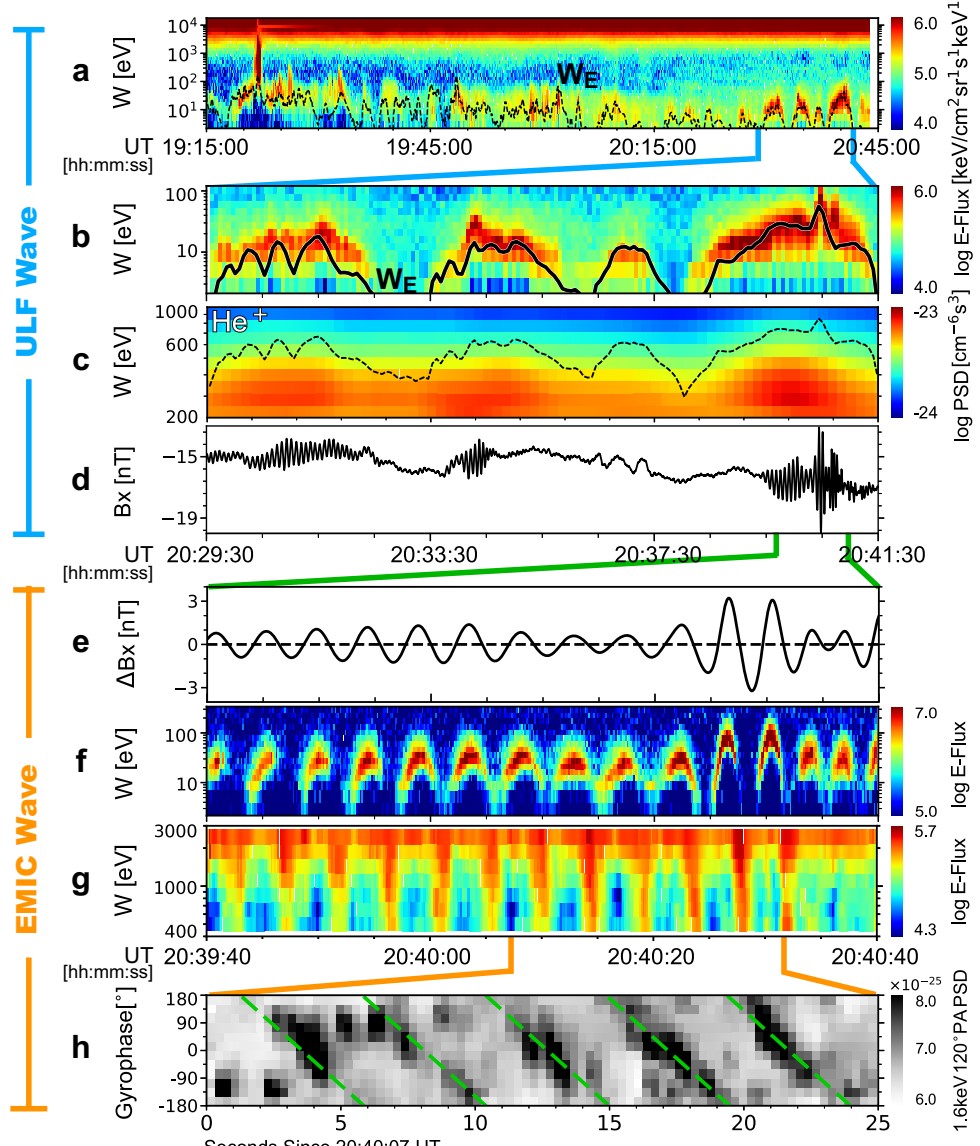

**Fig. 2 | Cross-scale interactions of ULF waves, EMIC waves and He⁺ ions in the January 7, 2019, event.** Fast mode data is used in panels a and b, fast survey mode data is used in panel c, and burst mode data is used in panels f-h. (**a**) The energy-time spectrogram of ion spin-averaged energy-fluxes measured by FPI. The black curve corresponds to $W_{E,H^+}$. (**b**) An expanded view of panel a. (**c**) He⁺ PSDs from HPCA. Oscillations can be observed, as indicated by the black curve representing $15W_{E,He^+}$. (**d**) The GSE-X component of the magnetic fields. (**e**) The EMIC wave magnetic fields in the DBCS coordinates. (**f** and **g**) The FPI energy-fluxes of cold ions and **h**ot ions, respectively. (**h**) A gyrophase-time spectrogram of the PSDs of 1.5-1.8 keV, 105°-135° PA ions from FPI. The green lines are defined according to the gyrophase of $-\mathbf{B_{w,EMIC}}$.

Figure 1a shows the magnetic field in this event. Clearly, the field variation can be considered as a superposition of two waves with very different periods, about 3.4 min and 7.5 s. We first focus on the longer-period one, which falls into the ULF range[6]. Figure 1c shows the magnetic field of the ULF waves in a field-aligned coordinate (FAC) system. Here, a bandpass filter (0.25–7 min) is used when generating this panel. We note that the parallel component of the ULF waves is very significant and even a little larger than the radial and azimuthal components, indicating the ULF waves are possibly compressional mode (e.g., mirror-mode structures[7,8]). However, unlike other mirror-mode-like structures reported in the literature[7,8], the parallel component of the ULF waves observed here is not dominant; the amplitude of two transverse components is almost equal to that of the parallel component. In addition, the observed electric fields also oscillate quasi-periodically with a period of about 3.4 min, as shown in Fig. 1d, e (as expected, the parallel component of the electric field is almost zero,

see Supplementary Fig. 1a). As we will show below, these electric-field variations result in energy flow between the ULF waves and nearby ions.

Besides the ULF waves, Fig. 1a also shows four packets of EMIC waves. The dynamic spectra (Fig. 1i) indicate that these EMIC waves are in the hydrogen band. Namely, their frequency (0.18 Hz or $0.36f_{cp}$, where $f_{cp} \approx 0.5$ Hz denotes the gyro-frequency of H⁺ ions) lies between H⁺ and helium (He⁺) ion gyro-frequencies. A remarkable feature noted here is that the EMIC-wave power (Fig. 1h) correlate well with the ULF waves. The correlation coefficient (CC), derived from a cross-wavelet analysis[9], is about 0.9 during the periods of the ULF waves (Fig. 1j).

In the light of the correlation, we suggest that the ULF waves are responsible for the periodicity of the EMIC-wave packets. As suggested by previous studies[10–12], magnetospheric EMIC waves are usually generated by the ion cyclotron anisotropy (ICA) instability, in which hot anisotropic ions act as free energy sources. This is the case here. When

interacting with the ULF waves, H$^+$ ions are quasi-periodically accelerated, especially those in the energy range of 9.8–26.6 keV (Fig. 1b). As a result, their velocity distributions quasi-periodically become more dominated by perpendicular motion than field-aligned motion, or in short, become more anisotropic (Fig. 1g). These quasi-periodic anisotropy enhancements coincide with the EMIC-wave packets. A one-to-one correspondence can be identified between them, if one compares Fig. 1g, h. Further, we note that when EMIC waves appear, the 9.8–26.6 keV H$^+$ ions are extracting energy from the ULF waves. Figure 1f shows the ion energy gain calculated from the observed ULF-wave electric fields ($\mathbf{E_{w,ULF}}$) and ion velocity ($\mathbf{v_{i,ULF}}$) distributions, $\frac{dW}{dt} = e\mathbf{E_{w,ULF}}\mathbf{v_{i,ULF}}$. One can see that $\frac{dW}{dt}$ is positive when EMIC waves are significant. These positive values mainly result from the radial component of $\mathbf{E_{w,ULF}}$ and $\mathbf{v_{i,ULF}}$, which, as shown in Fig. 1e, are highly correlated and approximately in phase during the whole time period of interest (CC = 0.76 and phase shift = 24°; see Supplementary Fig. 1c and 1d). On the other hand, the azimuthal component of $\mathbf{E_{w,ULF}}$ and $\mathbf{v_{i,ULF}}$ are not well correlated (CC < 0.5). Also, their contribution to the $\frac{dW}{dt}$ is smaller than that of the radial component (Supplementary Fig. 1e).

The above observations lead us to a two-step scenario for the generation of the EMIC waves: First, the ULF waves quasi-periodically accelerate H$^+$ ions and increase their anisotropy, and then, the resulting anisotropic H$^+$ ions quasi-periodically excite/amplify the EMIC waves via the ICA instability. This scenario is consistent with the cyclotron-resonance condition[10–12]. This condition is met when the Doppler-shifted EMIC wave frequency seen by ions equals ion gyro-frequency, which corresponds to a mathematical condition $\omega - k_\parallel v_\parallel = \Omega$, where $\omega$ and $k_\parallel$ represent the wave angular frequency and parallel wavenumber of the EMIC waves, and $\Omega$ and $v_\parallel$ the ion angular gyro-frequency and parallel velocity. For the EMIC waves considered here, the minimum resonant energy given by the resonance condition, $\frac{1}{2}m\left(\frac{\omega - \Omega}{k_\parallel}\right)^2$ where $m$ represents ion mass, is about 1.6 keV (the cold plasma dispersion relation[2] is used; see Methods, subsection estimation of the EMIC wave phase speed), lower than the energy of the H$^+$ ions considered here. Hence, these H$^+$ ions indeed can resonate with and provide free energy for the observed EMIC waves.

It is noted that the mechanism suggested here does not require any specific sources of the ULF waves. Especially, it can happen even when ULF waves are internally excited by ions[7,13–15], since in this situation, ULF waves can still periodically transfer energy to H$^+$ ions and increase their anisotropy, enabling them to excite EMIC waves. During the growth of EMIC waves, part energy flows to EMIC waves and does not return to ULF waves anymore. In this way, energy transfer proceeds from macroscales to microscales, although in this case, the ultimate energy source is ions and ULF waves only act as an intermediate step.

To fully elucidate the cross-scale wave–particle interactions, it is crucial to show how ions exchange energy with EMIC waves. However, the lack of burst mode data in this event prevents us from investigating this further. Fortunately, we have found a cross-scale interaction event in which burst mode data is available. We turn to this event in the rest of the paper.

### Overview of the second event

The second event was observed on January 7, 2019, which was also a quiet day in terms of the geomagnetic index (Dst≈0 nT and AE < 300 nT). In this event, MMS was located in the outer duskside magnetosphere (GSE [6.2, 7.8, 1.0] Earth radius, L-shell = 10.3 and magnetic local time = 15.1 h), and approximately 9.8° south of the magnetic equator. This event has been reported previously for studying cold (<100 eV) H$^+$ ion motion in EMIC waves[16]. Here, we investigate how cross-scale wave–particle interactions control energy flow in this event. As an interesting feature, we point out ahead that the particles involved here are He$^+$ ions, a minor species of the Earth's magnetosphere. Thus, this

event suggests that even minor species can also mediate cross-scale wave–particle interactions and the consequent cross-scale energy transfer processes.

### ULF-wave–ion interactions

The ULF waves were observed between 19:15-20:45 UT. Together with oscillations in fields, MMS also observed modulations in ion energy-fluxes. Figure 2a shows the energy-time spectrogram of the look-direction averaged ion energy-fluxes. The series of bridge-like arcs in the spectrogram is coincident with the quasi-periodic variations in ion bulk flow velocity. These ion bulk flows oscillate at the ULF-wave period as they undergo $E \times B$ drift motion ($\mathbf{V_E} = (\mathbf{E_{w,ULF}} \times \mathbf{B})/B^2$, where $\mathbf{B}$ represents the total magnetic fields) in response to the ULF-wave electric fields[17,18]. The ions thus acquire a kinetic energy $W_E = \frac{1}{2}m|\mathbf{V_E}|^2$. The ULF-wave fields oscillate in time, and consequently, the ions suffer bulk acceleration and deceleration quasi-periodically, which manifests as the bridge-like arcs[18] (termed $E \times B$-drift arcs hereinafter) in Fig. 2a.

Figure 2b shows an expanded view of the interval 20:29:30-20:41:30 UT. The three $E \times B$-drift arcs confirm that ULF-wave bulk-ion acceleration is taking place. This acceleration is perpendicular to the magnetic field and results in ion phase space densities (PSDs) that are periodically larger for perpendicular motion than for field-aligned motion, as shown in Supplementary Fig. 2a, b. The ion PSDs are dominated by field-aligned motion when $W_E$ is zero instantaneously during each wave period, a known characteristic of ionospheric outflow[19,20]. Consequently, the ion PSDs at MMS comprise plasma from the ionosphere. The plasmasphere or any associated plume is not the ion source because it would require ambient PSDs dominated by perpendicular motion[18–20].

Besides cold ions, the ULF waves also interact with hot He$^+$ ions in the energy range of 200–600 eV. Figure 2c shows the PSDs of these ions. Coincident with the $E \times B$-drift arcs shown in Fig. 2b, the PSDs also show quasi-periodic enhancements. The enhancements are strongest at about 400 eV, an energy much greater than the $E \times B$ drift energy of He$^+$ ions (generally <100 eV). Further, based on an analysis of He$^+$ ion energy spectrum, the contributions from $E \times B$ drift to the PSD enhancements are estimated to be less than 20% (see Supplementary Methods, subsection treatment of ion measurements). Hence, instead of $E \times B$ drift, the observed PSD enhancements should be caused by another type of ULF-wave–particle interactions—drift–bounce interactions[17,21,22]. This type of interactions occur between ULF waves and particles' drift and bounce motion, and would preferentially accelerate particles in the perpendicular direction. The efficiency of the interactions peaks when they take the form of resonance. However, because of the complex configuration of the dayside outer magnetosphere (e.g., the existence of off-equatorial magnetic minima[23]), we cannot either confirm or rule out unambiguously the occurrence of resonance in this event. Nevertheless, this does not affect our analysis much, since what is essential here—local energy flow from ULF waves to He$^+$ ions and ULF-wave-induced enhancements in He$^+$-ion anisotropy—has been directly observed (shown below).

### ULF-wave-EMIC wave coupling

Similar to the first event, the ULF waves in the second event are coupled with EMIC waves. As shown in Fig. 2d (also see Supplementary Fig. 3), three wave packets are observed during 20:29:30-20:41:30 UT. The frequency of these waves, about 0.22 Hz (0.3$f_{cp}$), lies between the H$^+$ and He$^+$ ion gyro-frequencies (Supplementary Fig. 3c). A singular value decomposition analysis reveals that the 0.22 Hz waves are left-hand circularly polarized and propagating anti-parallel to the background magnetic field[16]. Hence, the 0.22 Hz waves fit the classification reserved for EMIC waves. The EMIC wave packets are correlated with the observed ULF waves, with a cross-wavelet CC of 0.7 (Supplementary Fig. 3d). Moreover, a direct comparison between Fig. 2c, d indicates that the EMIC wave packets emerge when He$^+$ ion PSDs increase,

and correspondingly, disappear when the He$^+$ PSDs decay. The correlation between the ULF and EMIC waves indicates that the growth and damping of microscale (~$10^3$ km) EMIC waves are controlled by the macroscale (~$10^5$ km) influence of ULF waves. However, we note that this good correlation does not necessarily indicate that the EMIC waves are generated locally; the anti-parallel propagation of the EMIC waves indicates they are more likely generated at a higher latitude[24] and then amplified near MMS by ULF waves. Nevertheless, distinguishing local generation and amplification does not affect our conclusions, since as shown in what follows, local energy flow responsible for the growth of the EMIC waves has been directly measured.

From an instability point of view, the ULF-wave-EMIC wave coupling is primarily realized via the ICA instability. As shown in Supplementary Fig. 2d, e, the anisotropy of 200–600 eV He$^+$ ions increases in rhythm with the ULF waves, and becomes large and positive when EMIC waves are observed (see Supplementary Fig. 4 for corresponding velocity distributions). In contrast, the He$^+$ anisotropy is negative, when EMIC waves are absent. Besides, calculation suggests that the energy flow is directed from ULF waves to ions when large-amplitude EMIC waves are observed (Supplementary Fig. 2c). Hence, this event can also be understood with the aforementioned two-step scenario: The ULF waves first quasi-periodically accelerate hot He$^+$ ions and increase their anisotropy, and then, the anisotropic He$^+$ ions quasi-periodically excite/amplify the EMIC waves via the ICA instability.

As shown in what follows, gyrophase bunching indicative of strong nonlinearity is observed in this event. Hence, it is not appropriate to apply any linear or quasi-linear instability analyses (e.g., refs. 10,11) here. To reasonably describe the nonlinear ICA instability, hybrid or particle-in-cell numerical simulations are needed in general, which are much beyond our scope here. On the other hand, the local energy flow for EMIC wave growth is observed here directly and is sufficiently large. Thus, unlike in previous studies without direct observations, here we can conclude from observations themselves that the hot anisotropic He$^+$ ions observed can provide enough free energy for the growth of the EMIC waves.

## EMIC wave–ion interactions

The EMIC waves-ion interactions were observed during 20:39:40-20:40:40 UT when instruments on MMS were in burst mode. Figure 2e presents the $x$ component of the EMIC-wave magnetic field ($\mathbf{B}_{w,EMIC}$) in a de-spun body coordinate system (DBCS) (also see Supplementary Fig. 5), while Fig. 2f shows energy-fluxes of cold ions moving along the same direction. The energy fluxes oscillate in phase with $B_{w,EMIC}$. Here, the cold ions are dominated by H$^+$ ions, which have a gyro-frequency (0.7 Hz) three times larger than the EMIC wave (0.22 Hz). Therefore, as for ULF waves, the motion of cold H$^+$ ions in the EMIC wave fields can be approximated by their $E \times B$ drift[16], as supported by the observations of ion bulk velocity that are nearly equal to the calculated $E \times B$ drift velocity (Supplementary Fig. 5d). In this event, the $E \times B$ drift direction is along $\mathbf{B}_{w,EMIC}$ because of the anti-parallel propagation of the EMIC waves. Consequently, cold ions rotate in-phase with $\mathbf{B}_{w,EMIC}$, manifesting as in-phase stripes on a spectrogram like Fig. 2f.

As was the case for cold ions, energy-fluxes of hot ions also oscillate periodically (Fig. 2g). This oscillation corresponds to a group of ions bunched by and rotate with the EMIC waves. As an example, Fig. 2h shows the gyrophase distributions of 1.65 keV, 105°-135° PA ions within 20:40:07-20:40:32 UT, when the EMIC waves reach their maximum amplitude and then decay. The gyrophase distributions feature a series of inclined stripes, suggesting that at a particular time, ions are locked to a specific gyrophase rather than distributed uniformly. The stripes generally extend from the left-top to the bottom-right and appear every 4.5 s, as suggested by the green lines defined according to the gyrophase of $-\mathbf{B}_{w,EMIC}$. All these suggest that the observed gyro-anisotropic distributions are a consequence of the EMIC waves that

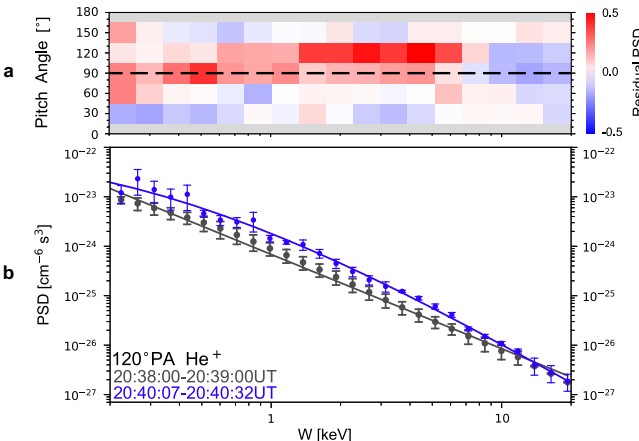

**Fig. 3 | EMIC wave–He$^+$ ion interactions. a** The PA-energy spectrogram of ion residual PSDs defined as $PSD_w/PSD_{bg} - 1$, where $PSD_{bg}$ and $PSD_w$ represent PSDs just before (20:38:00-20:39:00 UT) and during (20:40:07-20:40:32 UT) the EMIC waves, respectively. FPI burst mode data is used here. **b** The energy spectra of He$^+$ ions before (black dots) and during (blue dots) the EMIC waves. The curves are used to guide the eyes. The HPCA fast survey mode data is used here. Definitions of the error bars are given in Supplementary Methods, subsection treatment of ion measurements.

also left-hand rotate every 4.5 s. The 1.65 keV, 105°-135° PA ions are phase bunched by the EMIC waves.

Besides temporally rotating about the magnetic fields, the PSD distributions of hot ions also experience net variations. Figure 3a shows residual PSDs. The residual PSDs are positive at about 1 keV and 105°-135° PA, indicating enhancements in PSDs here. The PSD enhancements are secular rather than oscillatory, since what is shown here is the average over more than five wave cycles. It is noted that the PSD increases should be attributed to He$^+$ ions rather than H$^+$ ions, since the HPCA instruments reveals similar increases in He$^+$ ion PSDs but no corresponding variation in H$^+$ ion PSDs (Supplementary Fig. 6). Figure 3b presents the PSD energy spectra of 105°-135° PA He$^+$ ions. Black and blue dots denote spectra before and during the EMIC waves, respectively. As illustrated by the blue curve in Fig. 3b, enhancements in He$^+$ ion PSDs in the relevant energy range are observed when EMIC waves are significantly enhanced in amplitude over background values.

Phase bunching is observed at all energies where secular PSD increases occur and not only at 1.65 keV. Figure 4a shows the normalized gyrophase distributions of 0.34-6.19 keV 105°-135° PA ions. These spectrograms are derived from a superposed epoch analysis for five wave cycles during 20:40:07-20:40:32 UT (see Supplementary Methods, subsection treatment of ion measurements). The beginning of each wave cycle is chosen specifically such that dashed black lines in the figures represent the gyrophases of $-\mathbf{B}_{w,EMIC}$. In each spectrogram, a red stripe representing larger PSD appears at the gyrophases of $-\mathbf{B}_{w,EMIC}$. This striping suggests the observed gyrophase distributions are phase bunched by the EMIC waves at these energies and PAs. On the other hand, phase bunching is barely registered at other energies and PAs (Supplementary Fig. 7) where secular PSD increases are either very weak or non-existent. The one-to-one correspondence between phase bunching striping and secular PSD increases indicates the phase bunching should also be attributed to He$^+$ ions rather than H$^+$ ions. This conclusion is further supported by the HPCA observations that suggest the PSDs of He$^+$ ions at about 1 keV are roughly three times larger than that of H$^+$ ions when phase bunching occurs, even though the total number density of He$^+$ ions (0.04 cm$^{-3}$) is only 2% of that of H$^+$ ions (2 cm$^{-3}$). Hence, the 1 keV ions detected by FPI are constituted mainly by He$^+$ ions.

It is worth noting that the occurrence of the EMIC wave-He$^+$ ion interaction follows from the cyclotron-resonance condition. For the

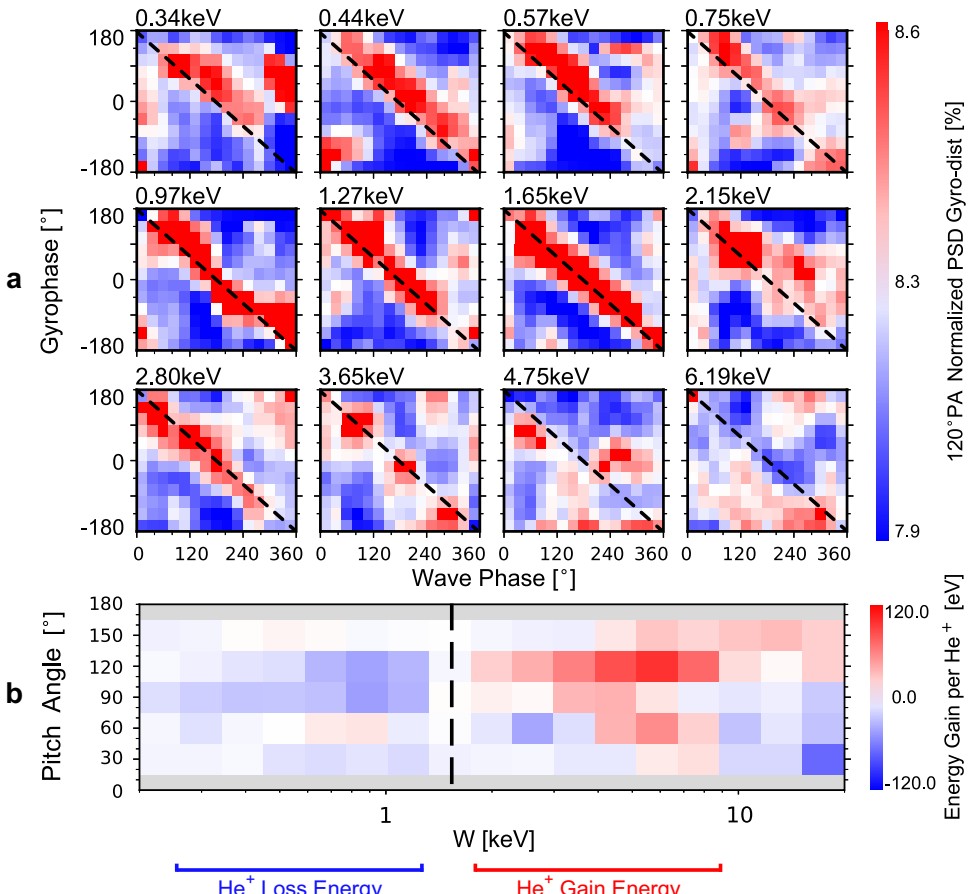

**Fig. 4 | Phase-bunching striping and He$^+$-ion energy gain in the rest frame of the plasma.** FPI burst mode data is used here. **a** The gyrophase-wave phase spectrograms of the normalized ion PSDs, derived from a superposed epoch analysis on the FPI measurements during 20:40:07-20:40:32 UT. Here, the distributions are normalized to make that, at any given wave phase, the sum of the normalized PSDs is a unit. The black lines represent the gyrophase of $-\mathbf{B}_{\mathbf{w,EMIC}}$. **b** The gyrophase-averaged energy gain per ion during 20:39:40-20:40:40 UT. Here we assume that all ions detected by FPI are constituted by He$^+$ in the energy range considered.

EMIC waves considered here, the parallel phase speed, $\omega/k_\parallel$, is about 590 km/s according to the cold plasma dispersion relation. Thus, in the PA range of 105°-135°, and for He$^+$ ions, the resonant energy, $\frac{1}{2}m\left(\frac{\omega-\Omega}{k_\parallel\cos PA}\right)^2$, is between 0.6 and 4.5 keV. These estimates match the phase bunching seen by MMS, suggesting it results from EMIC wave-He$^+$ ion cyclotron resonance.

Further analysis of Fig. 4a reveals that the He$^+$ phase-bunching stripes are not exactly coincident with the black lines representing the gyrophase of $-\mathbf{B}_{\mathbf{w,EMIC}}$. This feature suggests a net energy exchange between the EMIC waves and He$^+$ ions. Figure 4b shows the gyrophase-averaged energy gain per He$^+$ ion (see Methods, subsection energy transfer in EMIC wave–ion cyclotron resonance.). On the one hand, the energy gain of 0.2–1 keV He$^+$ ions is negative, implying this ion population feeds energy to the EMIC waves. Integrating over the phase space of He$^+$ ions, the total rate of energy flow from the 0.2–1 keV He$^+$ population to the EMIC waves is 0.51 eV/cm$^3$s. Using the observed $|\mathbf{B}_{\mathbf{w,EMIC}}|$ of about 3 nT, the EMIC wave energy density, $\sim\frac{|\mathbf{B}_{\mathbf{w,EMIC}}|^2}{2\mu_0}$, is 22 eV/cm$^3$, so that roughly ten wave periods (about 45 s) are needed to accumulate this wave energy density, indicating that the He$^+$ population has sufficient free energy to excite/amplify the EMIC waves. On the other hand, the energy gain of 1–10 keV He$^+$ ions is positive, implying the EMIC waves transfer energy to and accelerate this He$^+$ population. Combine the negative and positive energy flow, the net rate of energy flow to EMIC waves from all available He$^+$ ions is still positive, on the order of about 0.24 eV/cm$^3$s.

Figure 5 summarizes the energy transfer processes in the second event. Macroscale (~10$^5$ km) ULF waves first accelerate He$^+$ ions of hundred eVs and increase their anisotropy. Then, the anisotropic He$^+$ ions transfer energy to microscale (~10$^3$ km) EMIC waves, leading to the control of the EMIC waves by the ULF waves. Finally, the EMIC waves accelerate >1 keV He$^+$ ions, causing further ion energization and final energy dissipation.

## Discussions

In summary, the observations presented here demonstrate that ULF waves can also modulate and amplify EMIC waves by providing additional free energy, besides lowering the thresholds of EMIC wave instabilities by reducing the local magnetic field strength as suggested by previous studies[25]. In a broader context, our results present observational evidence for the importance of cross-scale wave–particle interactions in collisionless plasmas. First, the observations quantitatively confirm that cross-scale wave–particle interactions can lead to efficient energy transfer from macroscales to microscales, providing us with another mechanism for cross-scale energy transfer processes besides the turbulent local cascade model. Second, the observations suggest that cross-scale wave–particle interactions can expand the energy range of particles involved in wave–particle energy exchange processes and consequently might lead to more efficient plasma energization. As shown in the second event, only ~10$^2$ eV He$^+$ ions can be energized by the ULF waves, if there were no cross-scale wave–particle interactions. However, with cross-scale wave–particle interactions, He$^+$ ions of ~10$^3$ eV are also accelerated. As a closing

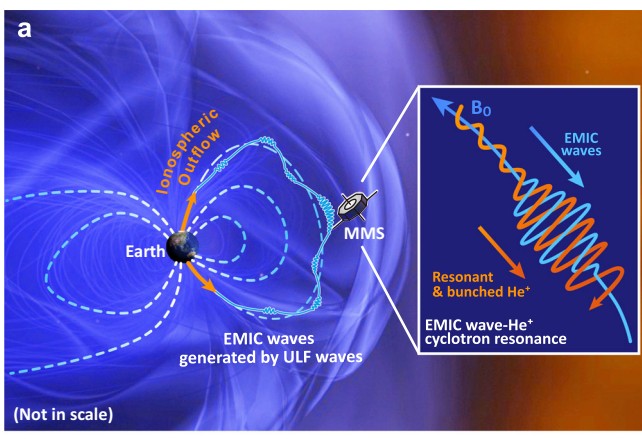

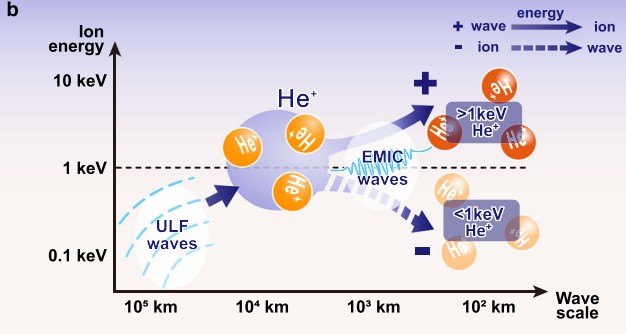

**Fig. 5 | Schematic diagrams summarizing the simultaneous macroscale and microscale wave–ion interactions observed on January 7, 2019. a** The coupling between macroscale ULF waves and microscale EMIC waves. The insert illustrates the cyclotron resonance between the EMIC waves and He$^+$ ions. **b** Energy flow in this event. The energy first transfers from ULF waves to several hundred eV He$^+$, then from He$^+$ to EMIC waves, and finally from the EMIC waves to He$^+$ of keV. The simultaneous macroscale and microscale wave–ion interacting can cause efficient cross-scale energy transfer and plasma energization in astrophysical and space plasmas.

remark, we would like to point out that cross-scale wave–particle interactions can also occur in other astrophysical and space plasma systems as long as there is more than one type of plasma waves of different scale, such that they can simultaneously interact with the same particle population. The events analyzed here are examples of this more general mechanism.

## Methods

### Coordinate systems

Most data in this study is presented in the Geocentric Solar Ecliptic (GSE) coordinate system, with the x-axis pointing towards the Sun along the Sun-Earth line, the z-axis oriented along the ecliptic north pole, and the y-axis completing the right-handed coordinate system. When presenting field and ion bulk velocity, a field-aligned coordinate (FAC) system is also used, where, by convention, the parallel axis (denoted as p) is defined as the direction of the local background magnetic field (averaged over the time interval of corresponding figures), the azimuthal axis (denoted as a) is perpendicular to both the spacecraft position vector and the parallel axis (increases eastward), and the radial axis (denoted as r) completes the right-handed coordinate system. The de-spun coordinate system (DBCS) is used in Fig. 2, with the z-axis aligned with the spacecraft's spin axis, the y-axis perpendicular to both the DBCS z-axis and the Sun-spacecraft line (positive in the direction from the dawn to the dusk), and the x-axis completing the right-handed coordinate system. Since the spin axis of MMS is approximately along the GSE z-axis in the corresponding time intervals, the DBCS can be considered near the GSE coordinate system.

### Estimation of the EMIC wave phase speed

The dispersion relation under cold plasma assumption[2] is used to estimate the phase speed ($v_{\phi,\text{EMIC}}$) of the EMIC waves:

$$\frac{c^2}{v_{\phi,\text{EMIC}}^2} = \frac{c^2 k_\parallel^2}{\omega^2} = 1 - \frac{\omega_{\text{pp}}^2}{\omega\left(\omega - \Omega_{\text{p}}\right)} - \frac{\omega_{\text{pe}}^2}{\omega\left(\omega + \Omega_{\text{e}}\right)} \qquad (1)$$

where $\omega_{\text{pp}}$ and $\omega_{\text{pe}}$ are the plasma frequencies of protons and electrons, $\Omega_{\text{p}}$ and $\Omega_{\text{e}}$ are the gyro-frequency of protons and electrons, respectively, $c$ is the light speed, and $\omega$ and $k_\parallel$ are the angular frequency and parallel wavenumber of the waves, respectively. Here, as a first order approximation, the effects of heavy ions on the dispersion relation are neglected, since the total number density of He$^+$ and O$^+$ ions are only about 2 and 1% of that of H$^+$ ions, respectively. In the second event, the plasma number density is 2.2 cm$^{-3}$ and the strength of the background magnetic field is 46 nT. Accordingly, $v_{\phi,\text{EMIC}}$ is 590 km/s, corresponding to a wavelength of about 2700 km.

### The spatial scale of waves

In this paper, wavelength is used as a proxy for the spatial scale of waves. For EMIC waves, we first derived phase speed from the dispersion relation given above and then calculated wavelength according to $\lambda_{\text{EMIC}} = v_{\phi,\text{EMIC}} T_{\text{EMIC}}$, where $\lambda_{\text{EMIC}}$ and $T_{\text{EMIC}}$ represent the wavelength and period, respectively. The result suggests that the spatial scale of the observed EMIC waves is ~$10^3$ km. For ULF waves, wavelength is estimated from the dispersion relation for Alfven waves, $\omega = v_A k$, where $\omega = 2\pi/T_{\text{ULF}}$, $k = 2\pi/\lambda_{\text{ULF}}$, and $v_A = B_0/\sqrt{\mu_0 m n}$ denotes the Alfven velocity. Given the measured magnetic fields and plasma density, the wavelength, or the spatial scale of the observed ULF waves, is ~$10^5$ km.

It is noted that the obtained wavelength of both the ULF waves and EMIC waves is much larger than the MMS spacecraft separation (approximately 25 km). This, together with the fact that the magnetic fields observed by the four MMS spacecraft are almost the same (Supplementary Fig. 5a), suggests that the ULF waves and EMIC waves are approximately homogenous on the spacecraft separation. Therefore, to improve data quality, data (both field and particle data) averaged across all four MMS spacecraft are used in this paper.

### Working definition of the rest frame of the plasma

The rest frame of the plasma is used when calculating the anisotropy of H$^+$ and He$^+$ ions (Supplementary Figs. 2, 4), and the energy exchange between He$^+$ ions and EMIC waves (Fig. 4b). The working definitions of these frames are:

Figure 4b. Here, the rest frame of the plasma is determined from the He$^+$ ion bulk velocity given in the HPCA fast survey mode data. We first applied a low-pass filter with an upper cutoff frequency of 0.05 Hz (six times less than the frequency of the EMIC waves) to the bulk velocity during 20:39:40-20:40:40 UT. Then, the resulting bulk velocity was linearly interpolated onto the epoch series of FPI burst mode data. The obtained bulk velocity series is then used to calculate the rest frame of the plasma. (As determined in this way, there is no ULF electric field in this frame of reference.)

Supplementary Fig. 2. For Supplementary Fig. 2d, e presenting He$^+$ ion anisotropy, the rest frame of the plasma is determined according to the bulk velocity of He$^+$ ions given in the HPCA fast survey mode data. For Supplementary Fig. 2f showing H$^+$ ions, the rest frame of the plasma is determined according to the bulk velocity of H$^+$ ions given in the HPCA fast survey mode data.

Supplementary Fig. 4. Here, the rest frame of the plasma is defined according to the He$^+$ ion bulk velocity given in the HPCA fast survey mode data. The frame transformation is applied prior to the time average.

## Energy transfer in EMIC wave–ion cyclotron resonance

Figure 4b shows the gyrophase-averaged energy gain per $He^+$ ion in the rest frame of the plasma, which is defined as

$$\int_{t1}^{t2} <e\mathbf{E_{w,EMIC}v_i}>_g dt \tag{2}$$

where $\mathbf{E_{w,EMIC}}$ denotes the EMIC wave electric fields, $e$ electron charge, and $\mathbf{v_i}$ the velocity of $He^+$ ions (all the variables are in the rest frame of the plasma). The integral is taken over the interval 20:39:40-20:40:40 UT. The symbol, $<>_g$, denotes average over gyrophase, namely:

$$<e\mathbf{E_{w,EMIC}v_i}>_g = \frac{\int_0^{2\pi} (e\mathbf{E_{w,EMIC}v_i})\text{PSD}d\phi}{\int_0^{2\pi}\text{PSD}d\phi} \tag{3}$$

where PSD represents ion phase space density (assume all ions in the energy range of interest measured by FPI are $He^+$ ions), and $\phi$ represents gyrophase.

## Data availability

MMS data used in this study are archived at MMS Science Data Center (https://lasp.colorado.edu/mms/sdc/public), including magnetic field data (https://lasp.colorado.edu/mms/sdc/public/data/mms1/fgm/brst/l2/), electric field data (https://lasp.colorado.edu/mms/sdc/public/data/mms1/edp/brst/l2/dce/) and ion data (https://lasp.colorado.edu/mms/sdc/public/data/mms1/fpi/brst/l2/). The datasets generated during and/or analyzed during the current study are available from the corresponding author on reasonable request.

## Code availability

MMS data have been loaded, analyzed, and plotted using the SPEDAS software (Space Physics Environment Data Analysis Software). The SPEDAS software can be downloaded via the http://spedas.org/blog/ Downloads and Installation page.

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

## Acknowledgements

This work was supported by the Major Project of Chinese National Programs for Fundamental Research and Development 2021YFA0718600 (Q.G.Z.), the National Natural Science Foundation of China 42230202 (Q.G.Z) and the China Space Agency project D020301 (Q.G.Z). S.Y.F. appreciates the support from the National Natural Science Foundation of China (41731068). We thank the entire MMS team for providing high time-resolution field and ion data.

## Author contributions

Z.Y.L. conducted the study, analyzed the data and prepared the manuscript. Q.G.Z. supervised the study, contributed to data interpretation, and revised the manuscript. R.R., H.Z., X.Z.Z., S.Y.F., and C.Y. contributed to data analysis and revised the manuscript. Y.F.W and Y.Z.H. contributed to data analysis. C.J.P, S.A.F., and L.G contributed to data processing of FPI, HPCA, and FGM, respectively.

## Competing interests

The authors declare no competing interests.

## Additional information

Q.-G. Zong.

