## [Peer Review File · Nature Communications]

Reviewers' comments:

Reviewer #1 (Remarks to the Author):

This work is clearly of publishable standard and has been extensively reviewed and updated already. It does reveal evidence for an energy cascade between larger scale ULF waves and smaller scale EMIC waves through use of the high time resolution and small spatial scale of the MMS array. This cross-scale energy transfer is important as it enables a fast and efficient mechanism for plasma energization.

It seems the issue that has been pointed out is that although the measurements have unique accuracy, such cross-scale coupling processes have been shown (or conjectured) before, although it is not clear that previously such a complete and direct determination has been done (as is possible with the 4 spacecraft measurements). A second issue raised by referee3, however, is that the present paper follows the analysis of the same event as considered by Shi et al (2020), which undoubtedly it does, and indeed some aspects of the ion modulation by both EMIC and Pc5 (ULF) waves were pointed out by Shi et al.

The authors have refuted these claims and argue that their treatment is a substantial extension of that of Shi et al to show new results not considered there. I do accept that substantially more is done in the present paper, in particular on the wave coupling and amplification, but I feel this is still an extension and further interpretation of results already noted, albeit a significant one. It probably is for the editors to make a final decision as to whether this is substantial enough to warrant publication in *Nature Communications*. In case the answer is negative, however, I think one way the authors could provide an improvement that would potentially allow publication, would be to demonstrate one other event that shows that the same mechanism operates. In that case, I feel there would be little argument against publication in *NatComm*.

Referee1 points out that the importance of the cold ion population could be shown in other situations, and in response the authors admit that more events are needed, although they feel this study should inspire further analysis. Again, a second event would go a long way towards nailing

down this point. Most of the technical issues the referee raises appear to be answered by the authors.

Referee2 comments that other work has discussed some of these aspects and points out that Kitamura et al. has performed similar analysis with MMS using similar techniques. The authors claim that their analysis in the present paper produces unambiguous evidence for the mechanisms, but they don't explain why the previous studies were ambiguous using the same dataset. I feel that the present paper does connect some of the different aspects and interplay of instabilities in a more complete way than the previous studies have hinted at. However, I can only repeat that a second event showing the same balance of processes would be hugely convincing. Given the inspiring nature of this work, that might not be an onerous thing for the authors to try.

From my point of view, I accept that the present paper does build on previous work and has revealed a plausible scenario for how the coupling operates. However, I feel they would solidify this scenario by analysis of at least one other event. I feel the comments by the three referees are not unreasonable, but the authors have made efforts to make corrections to most of the technical queries. Whether these clarifications alone are sufficient to allow publication in Nature Communications is perhaps up to the editors.

Reviewer #2 (Remarks to the Author):

The authors have addressed most of my comments and expanded on their original analysis.

There are still a couple points that I find unclear. First, the Bx field measurements in Figure 1b do not show a correlation between EMIC waves packets and ULF wave phase; they look pretty random. There is no one-to-one correspondence between the ULF wave oscillations in panel 1b and the flux oscillations in panel 1c, either. What is the correlation coefficient between these time series? And what are the criteria for modulation applied here?

Second, the authors state that ULF waves quasi-periodically increase the PSD and anisotropy of 200-600 eV He⁺ ions and the resulting anisotropic He⁺ ions quasi-periodically amplify initially small amplitude EMIC waves via the ion cyclotron anisotropy instability.

However, it is not demonstrated using the linear theory whether the low-density 200-600 eV He⁺ suggested as the energy source provides enough free energy for positive EMIC wave growth. In other words, whether the He⁺ plasma beta and temperature anisotropy are close to or exceed the EMIC instability threshold. The instability analysis has been done by e.g., Kennel and Petschek, 1966 and Blum et al., 2009.

Overall, due to the similarities with other published studies (previously discussed), I believe this manuscript would be more suitable for a more specialized journal, for example, Geophysical Research Letters or Journal of Geophysical Research. The science advances presented here are rather incremental to justify publication in Nature Communications, though would be of interest to the magnetospheric community once the paper is revised.

References:

Blum, L. W., MacDonald, E. A., Gary, S. P., Thomsen, M. F., and Spence, H. E. (2009), Ion observations from geosynchronous orbit as a proxy for ion cyclotron wave growth during storm times, *J. Geophys. Res.*, 114, A10214, doi:10.1029/2009JA014396.

Reviewer #3 (Remarks to the Author):

This manuscript addresses interesting observations about cross-scale energy transfer in the magnetosphere using a unique dataset derived by the MMS spacecraft. I understand that the importance of the topic. However, additional justification is needed for acceptance. If it is difficult due to length constraints, it can be included in Supplemental Material. The comments are written below.

This manuscript addresses interesting observations about cross-scale energy transfer in the magnetosphere using a unique dataset derived by the MMS spacecraft. I understand that the importance of the topic. However, additional justification is needed for acceptance. If it is difficult due to length constraints, it can be included in Supplemental Material.

Comments

Comment 1

Lines 81-84: Although the energy of 400 eV appears to be much greater than the $E \times B$ drift energy, it needs to be discussed more carefully. When the $E \times B$ drift energy for H^+ is 25 eV, the $E \times B$ drift energy for He^+ becomes 100 eV, which corresponds to the velocity of ~ 70 km/s. In the $E \times B$ drifting frame, additional 70 km/s (30 km/s), which corresponds to the energy of 100 eV (20 eV) for He^+ , is needed to reach 400 eV (200 eV) in the spacecraft frame. Thus, the energy needed in the $E \times B$ drifting frame is much smaller than the energy seen by the spacecraft. Thus, I afraid that the high-energy tail of the warm He^+ distribution might affect the enhancement of PSD around a few hundreds of electron volts (seen in the spacecraft frame). Is it possible to rule out such a possibility? Do you think that the (perpendicular) temperature of He^+ (in the $E \times B$ drifting frame) was much lower than 100 eV (20 eV)? If so, the effect of high-energy tail may be small.

Comment 2

Lines 83-87: Is there any evidence that He^+ satisfied any drift-bounce resonance condition? If it is difficult to show it, is it possible to show existence of plausible resonance conditions that He^+ may be able to satisfy? If even it is difficult, is it possible to show any previous studies that such low-energy ions were accelerated by the drift-bounce resonance? I think that such discussion is necessary to reject the possibility mentioned in Comment 1. (As another possibility, the EMIC wave itself may heat He^+ in the perpendicular direction (non-locally?) and an excess energy may be returned to the waves at the position of the spacecraft. In such a case, ULF waves may not affect the energy transfer. It would be better to have some stronger evidence of the drift-bounce resonance to rule out such a possibility.)

Comment 3

Lines 107-110: If my understanding is correct, the “anisotropy” is calculated in the spacecraft rest frame. Anisotropy should be calculated in the plasma rest frame (\sim the $E \times B$ drifting frame). I afraid that the “anisotropy” was overestimated when $E \times B$ drift

velocity became large. Did the anisotropy in the plasma rest frame also fluctuate with the ULF wave? If this understanding is correct, “anisotropy” in the text and figures should be replaced by “flux ratio”. The “flux ratio” can be interpreted as anisotropy, only in cases when the $E \times B$ drift energy is negligible. This is the case for high-energy range and/or light ions (for example, hot H^+).

Comment 4

Supplementary Material, Lines 183-189: There are difficulties in calibration of the electric field data, especially the phase (and amplitude) of E_{wave} , because the response (phase rotation and sensitivity) of the electric field probes is affected by conditions of the surrounding plasma. Have you checked that the phase difference between B_{wave} and E_{wave} is 90° ? If there is an offset from 90° , how much does it affect the calculations of energy transfer, and have you checked to ensure that it does not change your conclusions? (About the amplitude, comparison between the estimated phase velocity and the ratio between E_{wave} and B_{wave} will be useful for validation.)

Comment 5

Supplementary Material, Lines 183-189: I understand that the He^+ velocity vectors are calculated in the spacecraft rest frame. Since this is the period when the $E \times B$ drift due to low-frequency (below the frequency of EMIC waves) electric fields is not small, it may be necessary to subtract the $E \times B$ drift velocity from the He^+ velocity vectors. As discussed in Comment 1, the difference in the frame has a large effect on the velocity vector (phase and magnitude) for the low-energy component. Since the component that was transferring energy to the EMIC wave was relatively low energy, any check seems to be necessary.

Minor comments

Comment 1

Lines 20, 21, 99, 100: What do the scales of the ULF and EMIC waves ($\sim 10^5$ and 10^3 km) mean? Since the wavelength of the EMIC wave was estimated as 2700 km, I think that they might mean the wavelength. If so, is it possible to say anything about the wavelength of the ULF wave from the observations? For example, Kitamura et al. (2021, *JGR*, <https://doi.org/10.1029/2020JA028912>) estimated a wavelength of ULF waves as ~ 3600 – 9000 km using the data obtained by the MMS spacecraft, although the wavelength is much smaller than 10^5 km. If the wavelength was very long, no difference must be

visible between the spacecraft. If so, I think that you can write that. Since the scales are important in this manuscript, it is good to define and demonstrate the scales as carefully as possible.

Comment 2

Line 50: Since this is a magnetospheric phenomenon, to show the spacecraft location in the magnetic latitude, magnetic local time, and L -value coordinate in addition to that in the GSE coordinates is useful for readers. Since it is a bit far from the Earth, I think that it is good to also add information about the distance and relative location (north or south) to minimum- B at low latitudes. (The location of minimum- B in the MMS MEC data may be usable.)

Comment 3

Lines 92-93: I suggest adding the wave frequency normalized by the cyclotron frequency of H^+ .

Comment 4

Lines 197-198: If my understanding is correct, “besides lower \sim ” is not come from the present result. To avoid misunderstandings, it would be better to emphasize more that it comes from other results.

Comment 5

Lines 216-217: What do the curves in Fig. 2b indicate?

Comment 6

Figures, Figures in Supplementary Material: There is no clear statement about the source of the data (one of the spacecraft or average of all spacecraft?). It is also better to describe the observation mode (fast survey or burst).

Comment 7

Supplementary Material, Line 26: For the calculation of He^+ anisotropy, what was used for the calculation as $f_{||}$? Any description is needed.

Comment 8

Supplementary Material, Fig. 5: What does the color scale mean? (What 0% and 100% mean?)

Comment 9

Supplementary Material, Lines 120-128: It is necessary to explain why you can ignore He^+ (and O^+).

Comment 10

Supplementary Material, Lines 120-128: To describe the minimum resonant energy for H^+ is useful for Supplementary Material Fig. 1e.

Comment 11

Supplementary Material, Lines 139-143: The description about FPI is incorrect. The energy range does not agree with Fig. 1a (2 eV?–20 keV?). The energy range does not appear to be log-evenly divided at least below 10 eV (Fig. 1b). The use of the two energy tables for the FPI burst data was supposed to have been stopped in 2016. Some description about the fast survey data is also necessary, if they are also used.

Comment 12

Supplementary Material, Lines 169-181: The energy and angular resolution of HPCA data (fast survey only?) should be described.

Correction

Supplementary Material, Line 24: 30° - 30° -> 30° - 150° ?

Supplementary Material, Line 40: Missing description about panel (b)

Supplementary Material, Line 137: 15 -> 150

Supplementary Material, Line 174: HPAC -> HPCA

Reviewer #1 (Remarks to the Author):

This work is clearly of publishable standard and has been extensively reviewed and updated already. It does reveal evidence for an energy cascade between larger scale ULF waves and smaller scale EMIC waves through use of the high time resolution and small spatial scale of the MMS array. This cross-scale energy transfer is important as it enables a fast and efficient mechanism for plasma energization.

We thank the reviewer for carefully reading the manuscript and making constructive comments. As suggested by the reviewer, we have revised the manuscript by adding a new event that shows the same mechanism operates. The observations are summarized in Fig. 1. Related discussions can be found in lines 59-100.

In the new event, the cross-scale energy transfer processes are caused by the cross-scale interactions of H^+ ions, ULF waves and EMIC waves. Here we chose an event in which H^+ ions are involved (rather than He^+ ions as in the old event), because we want to make a point that both the major and minor species of plasmas can mediate cross-scale wave-particle interactions and thus cross-scale energy transfer processes. Except for particle species, the overall mechanism in the new event is the same as the old event: First, the ULF waves (Fig. 1a) quasi-periodically accelerate 9.8-27 keV H^+ ions (Fig. 1b-c) and increase their anisotropy (Fig. 1g), and then, the resulting anisotropic H^+ ions quasi-periodically excite/amplify the EMIC waves (comparing Fig. 1g and Fig. 1f), leading to the control of the EMIC waves by the ULF waves (Fig. 1h).

Besides the new event described above, we have also found another six similar events (see Supplementary Table 1). Unfortunately, no burst mode data is available in all of these events (including the above one). As a result, we cannot analyze the energy transfer processes in these events in detail. It is very hard to find events in which cross-scale wave-particle interactions and burst mode data are simultaneously available. Actually, this is why we concentrate on the 2019/01/07 event, even though it has been reported before (but for very different purposes). Nevertheless, another seven events have been found, indicating cross-scale wave-particle interaction events are not rare. Future studies can provide us with more findings.

It seems the issue that has been pointed out is that although the measurements have unique accuracy, such cross-scale coupling processes have been shown (or conjectured) before, although it is not clear that previously such a complete and direct determination has been done (as is possible with the 4 spacecraft measurements). A second issue raised by referee3, however, is that the present paper follows the analysis of the same event as considered by Shi et al (2020), which undoubtedly it does, and indeed some aspects of the ion modulation by both EMIC and Pc5 (ULF) waves were pointed out by Shi et al.

We agree that such cross-scale coupling processes have been conjectured before. However, we suggest that our studies, for the first time, give direct, observational and quantitative

evidence for the processes and their importance (both with the four spacecraft measurements and the direct measurements of ion gyration).

We also would like to compare our studies with Shi et al. briefly here. Indeed, Shi et al. and our studies investigate the same event. However, the main focus of Shi et al. is the behavior (or more precisely, the $E \times B$ drift) of cold H^+ ions (<50 eV) in EMIC waves and ULF waves, whereas our studies concentrate on the resonant interactions of hot (>200 eV) He^+ ions, ULF waves and EMIC waves, and the resulting cross-scale energy transfer processes. None of our results is discussed in Shi et al.. Shi et al. and our studies investigate the same event but for entirely different purposes.

The authors have refuted these claims and argue that their treatment is a substantial extension of that of Shi et al to show new results not considered there. I do accept that substantially more is done in the present paper, in particular on the wave coupling and amplification, but I feel this is still an extension and further interpretation of results already noted, albeit a significant one. It probably is for the editors to make a final decision as to whether this is substantial enough to warrant publication in Nature Communications. In case the answer is negative, however, I think one way the authors could provide an improvement that would potentially allow publication, would be to demonstrate one other event that shows that the same mechanism operates. In that case, I feel there would be little argument against publication in NatComm.

Referee1 points out that the importance of the cold ion population could be shown in other situations, and in response the authors admit that more events are needed, although they feel this study should inspire further analysis. Again, a second event would go a long way towards nailing down this point. Most of the technical issues the referee raises appear to be answered by the authors.

Again, thanks very much for the comments. The manuscript has been revised accordingly. Please find the detailed response above.

Referee2 comments that other work has discussed some of these aspects and points out that Kitamura et al. has performed similar analysis with MMS using similar techniques. The authors claim that their analysis in the present paper produces unambiguous evidence for the mechanisms, but they don't explain why the previous studies were ambiguous using the same dataset. I feel that the present paper does connect some of the different aspects and interplay of instabilities in a more complete way than the previous studies have hinted at. However, I can only repeat that a second event showing the same balance of processes would be hugely convincing. Given the inspiring nature of this work, that might not be an onerous thing for the authors to try.

We agree that our paper gives a more complete investigation of the different aspects and interplay of instabilities. We sincerely appreciate your positive comment.

We did not mean that Kitamura et al. is ambiguous. This study is very clear and comprehensive. However, the main focus of Kitamura et al. is the EMIC wave-ion

cyclotron-resonance, whereas ours is cross-scale energy transfer processes mediated by cross-scale wave-particle interactions. Indeed, in terms of the EMIC wave-ion interactions, there are some overlaps between Kitamura et al. and our studies. However, we two focus on very different topics.

From my point of view, I accept that the present paper does build on previous work and has revealed a plausible scenario for how the coupling operates. However, I feel they would solidify this scenario by analysis of at least one other event. I feel the comments by the three referees are not unreasonable, but the authors have made efforts to make corrections to most of the technical queries. Whether these clarifications alone are sufficient to allow publication in Nature Communications is perhaps up to the editors.

We would like to express our gratitude to the reviewer again. The manuscript has been revised accordingly. Please find the details in the new version of the manuscript.

Reviewer #2 (Remarks to the Author):

The authors have addressed most of my comments and expanded on their original analysis.

We sincerely appreciate the constructive comments from the reviewer and have revised the manuscript accordingly. Please find our responses to the comments below.

There are still a couple points that I find unclear. First, the Bx field measurements in Figure 1b do not show a correlation between EMIC wave packets and ULF wave phase; they look pretty random. There is no one-to-one correspondence between the ULF wave oscillations in panel 1b and the flux oscillations in panel 1c, either. What is the correlation coefficient between these time series? And what are the criteria for modulation applied here?

Thanks very much for the comments. There is a good correlation between the EMIC wave packets and the ULF wave phase, and the oscillations in panel 2b and panel 2c (Please note that Fig. 2 in the revised manuscript corresponds to Fig. 1 in the old version).

For the correlation between the EMIC wave packets and the ULF wave phase. First of all, please note that the ULF waves in this event are most clearly manifested as electric field oscillations rather than magnetic field oscillations. This phenomenon, which is frequently reported in the literature, could be attributed to the odd harmonic standing wave structures of ULF waves, as the spacecraft are located near the magnetic equator in this event. Thus, it would be better to investigate the correlation between the EMIC wave packets and the ULF wave electric fields rather than magnetic fields. Supplementary Fig. 2d (and Fig. R1a attached below) shows the corresponding correlation coefficient, which is derived from a cross-wavelet analysis [Grinsted et al., 2004]. It is clear that a correlation coefficient as high as ~0.7 is reached at the periods of ~4-6 minutes, which is also approximately the periods of the ULF waves and EMIC wave packets. This correlation coefficient suggests that there is a good correlation between the EMIC wave packets and the ULF wave phase. This is also the criteria for modulation applied. Please find the related discussions in lines 156-158 in the revised manuscript.

For the second part of the comments. Fig. R1b attached below gives the cross-wavelet correlation coefficient between the ULF wave oscillations in Fig. 2b and the flux oscillations in Fig. 2c. We can see that, at the periods of ~4-6 minutes, the correlation coefficient is as high as ~1, indicating a good correlation between the two flux oscillations. Further, we also calculate the correlation coefficient between the ULF wave electric fields and the flux oscillations in Fig. 2c, and show the results in Fig. R2c. Again, a high correlation coefficient of ~0.8 is reached at the periods of ~4-6 minutes. These results conclude that there is a good correlation between the ULF wave oscillations in Fig. 2b and the flux oscillations in Fig. 2c.

Fig. R1. (a) The cross-wavelet correlation coefficient between the ULF wave E_y (Supplementary Fig. 1b) and the EMIC wave power at 0.22 Hz (Supplementary Fig. 1c). (b) The cross-wavelet correlation coefficient between the ~ 20 eV ion flux in Fig. 2b and the ~ 400 eV He+ PSDs in Fig. 2c. (c) The cross-wavelet correlation coefficient between the ULF wave E_y and the ~ 400 eV He+ PSDs.

Second, the authors state that ULF waves quasi-periodically increase the PSD and anisotropy of 200-600 eV He+ ions and the resulting anisotropic He+ ions quasi-periodically amplify initially small amplitude EMIC waves via the ion cyclotron anisotropy instability. However, it is not demonstrated using the linear theory whether the low-density 200-600 eV He+ suggested as the energy source provides enough free energy for positive EMIC wave growth. In other words, whether the He+ plasma beta and temperature anisotropy are close to or exceed the EMIC instability threshold. The instability analysis has been done by e.g., Kennel and Petschek, 1966 and Blum et al., 2009.

Thanks very much for the comments. The linear instability analysis has been done in the revised manuscript. Technically, we compared the EMIC anisotropy parameter ($\Sigma_{He} = \left(\frac{T_{\perp He}}{T_{\parallel He}} - 1\right) \beta_{\parallel He}^{\alpha_{He}}$) and the instability threshold parameter (S_{He}). According to the HPCA observations and Blum et al. [2009], we find $\beta_{\parallel He} = 0.04$, $\alpha_{He} = 0.46$ and thus $S_{He} = 0.12$. To obtain the parameter $\frac{T_{\perp He}}{T_{\parallel He}} - 1$, the observed velocity distribution functions of He+

ions are fitted to a bi-Maxwellian distribution (see Supplementary Fig. 3). Three time intervals corresponding to the three EMIC wave packets are shown. The parameter $\frac{T_{\perp He}}{T_{\parallel He}} - 1$ in the three time interval is 0.62, 0.50 and 0.84. Consequently, Σ_{He} is 0.14, 0.11 and 0.19. We can see that, Σ_{He} is greater than S_{He} in the first and third time intervals. In the second time interval, though Σ_{He} is a little smaller, it is still close to S_{He} . These results suggest that the hot He^+ ions considered indeed can provide enough free energy for positive EMIC wave growth. Discussions of this point have been added to the manuscript. Please find lines 176-184 in the revised manuscript.

Overall, due to the similarities with other published studies (previously discussed), I believe this manuscript would be more suitable for a more specialized journal, for example, Geophysical Research Letters or Journal of Geophysical Research. The science advances presented here are rather incremental to justify publication in Nature Communications, though would be of interest to the magnetospheric community once the paper is revised.

Again, we sincerely appreciate the referee for spending time reviewing this manuscript. Respectfully, we still suggest that our studies are very different from other publications (both Shi et al. and Kitamura et al.). We concentrate on cross-scale energy transfer processes mediated by cross-scale wave-particle interactions, whereas Shi et al. focus on cold H^+ ion behavior in EMIC waves and Kitamura et al. focus on EMIC wave-ion cyclotron resonance. There are indeed some overlaps, but our purposes are entirely different. In addition, we have made many further revisions to the new version of the manuscript. Especially, besides addressing all the technical issues, a new event that shows the same mechanism operates is included in the revised manuscript (Fig. 1 and lines 59-100). Therefore, based on the new expansions, we would like to ask if the reviewer could reconsider the general comment?

Reviewer #3 (Remarks to the Author):

This manuscript addresses interesting observations about cross-scale energy transfer in the magnetosphere using a unique dataset derived by the MMS spacecraft. I understand that the importance of the topic. However, additional justification is needed for acceptance. If it is difficult due to length constraints, it can be included in Supplemental Material.

We are very grateful to the reviewer for the constructive comments. We have given full consideration to the comments and revised manuscript thoroughly. Please find our detailed responses to the comments in the following letter.

Please note that we have included a new event in the revised manuscript (as suggested by reviewer #1). This event shows that the same mechanism can also operate for H^+ ions. The observations of the new event are summarized in Fig. 1 of the revised manuscript. Related discussions can be found in lines 59-100.

Major Comment 1

Lines 81-84: Although the energy of 400 eV appears to be much greater than the $E \times B$ drift energy, it needs to be discussed more carefully. When the $E \times B$ drift energy for H^+ is 25 eV, the $E \times B$ drift energy for He^+ becomes 100 eV, which corresponds to the velocity of ~ 70 km/s. In the $E \times B$ drifting frame, additional 70 km/s (30 km/s), which corresponds to the energy of 100 eV (20 eV) for He^+ , is needed to reach 400 eV (200 eV) in the spacecraft frame. Thus, the energy needed in the $E \times B$ drifting frame is much smaller than the energy seen by the spacecraft. Thus, I afraid that the high-energy tail of the warm He^+ distribution might affect the enhancement of PSD around a few hundreds of electron volts (seen in the spacecraft frame). Is it possible to rule out such a possibility? Do you think that the (perpendicular) temperature of He^+ (in the $E \times B$ drifting frame) was much lower than 100 eV (20 eV)? If so, the effect of high-energy tail may be small.

Thanks for the comments. The effects of the high-energy tail of the background warm He^+ ions are small, as suggested by an analysis of the observed energy spectrum. As an example, we show in Fig. R2 (attached below) the energy spectrum of 90° PA He^+ ions observed during 20:39:40-20:40:40 UT, when He^+ PSD enhancements and EMIC waves are observed (This is also the time interval primarily focused on in this study). Clearly, the energy spectrum is composed of two components: a power-law distribution below ~ 100 eV corresponding to the $E \times B$ -drifting background warm He^+ ions and a bump-on-tail distribution around 200-1000 eV corresponding to the PSD enhancements. By fitting the low-energy component to a power-law distribution and extending the resulting curve to 200-1000 eV, the contributions from the background warm He^+ ions to the PSD enhancements around 200-1000 eV are estimated to be less than 20%. Therefore, the high-energy tail of the background warm He^+ component does not affect the 200-1000 eV PSD enhancements much. The same conclusions can be reached for the other two He^+ PSD enhancements (20:29:10-20:30:10 and 20:34:30-20:35:30 UT).

We have included the above discussions in the revised manuscript. Please find lines 136-138 in the manuscript and lines 230-240 in the Supplementary Materials.

Fig. R2. The contributions from the background warm He⁺ ions to the PSD enhancements around 200-1000 eV.

Major Comment 2

Lines 83-87: Is there any evidence that He⁺ satisfied any drift-bounce resonance condition? If it is difficult to show it, is it possible to show existence of plausible resonance conditions that He⁺ may be able to satisfy? If even it is difficult, is it possible to show any previous studies that such low-energy ions were accelerated by the drift-bounce resonance? I think that such discussion is necessary to reject the possibility mentioned in Comment 1. (As another possibility, the EMIC wave itself may heat He⁺ in the perpendicular direction (non-locally?) and an excess energy may be returned to the waves at the position of the spacecraft. In such a case, ULF waves may not affect the energy transfer. It would be better to have some stronger evidence of the drift-bounce resonance to rule out such a possibility.)

Thanks very much for the helpful comments. First of all, we have to admit that there is no more direct evidence for the drift-bounce resonance between the ULF waves and the He⁺ ions. A major obstacle here is that we cannot accurately determine the azimuthal wavenumber (k_a) and the standing wave structures of the ULF waves, which is a problem frequently met in ULF wave-particle interaction studies. Please note that, in this event, the separation among the four MMS spacecraft is too small to accurately determine k_a via a timing method.

Nevertheless, we can show that there are possible resonance conditions that He⁺ ions are able to satisfy. The drift-bounce resonance conditions are given by $\omega - k_a \omega_d = N \omega_b$, where ω_b and ω_d are the bounce and drift frequency of He⁺ ions, ω and k_a are the angular frequency and azimuthal wavenumber of the ULF waves, and N is an integer representing the harmonics of the resonance (most likely to be within [-2, 2] according to previous studies). From the resonance conditions, we can estimate k_a and see whether it is possible. By calculating ω_b and ω_d for 200-1000 eV He⁺ ions in the T89 model (other models give very similar results), and taking ω to be $\sim \frac{2\pi}{6 \text{ min}}$ and N within [-2,2], we find k_a required by the resonance conditions is $\sim 80-7000$. We note that k_a within this range is frequently

reported in the literature: $k_a \sim 30, 17, 244$ and 314 in Le et al. [2021], ~ 216 and 261 in Yamamoto et al. [2019], ~ 200 in Takahashi et al. [2018], and ~ 100 in Min et al. [2017] (see the detailed references in the manuscript). Therefore, for He^+ ions considered here, the drift-bounce resonance is possible in terms of the resonance conditions. A discussion of this point has been included in lines 141-146 in the revised manuscript.

Further, we note that there are studies showing that such low-energy ions can be accelerated by drift-bounce resonances: e.g., Figure 3 of Zong et al. [2012] and Figure 1 of Wang et al. [2021].

Regarding the second part of this comment, we suggest that it is hard to apply the non-local EMIC wave-ion interaction scenario here. Please note that there is a good correlation between the He^+ PSD enhancements (Fig. 2c) and the ULF wave-induced $\mathbf{E} \times \mathbf{B}$ drift arcs (Fig. 2b). This good correlation suggests that, as the case for the latter, the former should also be caused by the ULF waves. The ULF wave-ion-EMIC wave interaction scenario proposed in our manuscript can give a comprehensive explanation of all the observations, which cannot be done easily by the non-local EMIC wave-ion interactions scenario, although it is possible in principle.

Major Comment 3

Lines 107-110: If my understanding is correct, the “anisotropy” is calculated in the spacecraft rest frame. Anisotropy should be calculated in the plasma rest frame (\sim the $\mathbf{E} \times \mathbf{B}$ drifting frame). I am afraid that the “anisotropy” was overestimated when $\mathbf{E} \times \mathbf{B}$ drift velocity became large. Did the anisotropy in the plasma rest frame also fluctuate with the ULF wave? If this understanding is correct, “anisotropy” in the text and figures should be replaced by “flux ratio”. The “flux ratio” can be interpreted as anisotropy, only in cases when the $\mathbf{E} \times \mathbf{B}$ drift energy is negligible. This is the case for high-energy range and/or light ions (for example, hot H^+).

Thanks very much for identifying this issue. Indeed, the “anisotropy” in the previous version is calculated in the rest frame of the spacecraft. Now, in the revised manuscript, the anisotropy is calculated in the rest frame of the plasma. As you can see in Supplementary Fig. 1, the new anisotropy is very similar to the old one and also oscillate with the ULF waves. Therefore, our conclusions still hold.

Major Comment 4

Supplementary Material, Lines 183-189: There are difficulties in calibration of the electric field data, especially the phase (and amplitude) of E_{wave} , because the response (phase rotation and sensitivity) of the electric field probes is affected by conditions of the surrounding plasma. Have you checked that the phase difference between B_{wave} and E_{wave} is 90° ? If there is an offset from 90° , how much does it affect the calculations of energy transfer, and have you checked to ensure that it does not change your conclusions? (About the amplitude, comparison between the estimated phase velocity and the ratio between E_{wave} and B_{wave} will be useful for validation.)

Thanks very much. In the revised manuscript, we have added a figure showing the phase

difference between B_w and E_w (Supplementary Fig. 4e). We can see that the phase difference is roughly $+90^\circ$, with the positive sign indicating anti-parallel propagation (as the waves are left-hand polarized). This result is consistent with our expectations. Supplementary Fig. 4f compares the amplitude of the observed electric fields (blue curve) and that calculated from $\vec{E}_w = \vec{v}_\phi \times \vec{B}_w$ (red curve). The observed curve roughly matches the theoretical one. The absolute and relative differences between them are ~ 0.3 mV/m and $\sim 25\%$, respectively. Therefore, we suggest that the electric field instruments performed well in this event, ensuring a reliable calculation of the energy transfer between plasma and wave fields.

We have added a brief discussion of this point in lines 129-141 in the Supplementary Materials.

Major Comment 5

Supplementary Material, Lines 183-189: I understand that the He+ velocity vectors are calculated in the spacecraft rest frame. Since this is the period when the $E \times B$ drift due to low-frequency (below the frequency of EMIC waves) electric fields is not small, it may be necessary to subtract the $E \times B$ drift velocity from the He+ velocity vectors. As discussed in Comment 1, the difference in the frame has a large effect on the velocity vector (phase and magnitude) for the low-energy component. Since the component that was transferring energy to the EMIC wave was relatively low energy, any check seems to be necessary.

Sorry for the confusion, but here the energy gains were calculated in the rest frame of the plasma. Therefore, the contributions from the $E \times B$ drift have already been removed. This point is stated more clearly in the revised manuscript. Please find line 318 in the main manuscript and line 243 in the Supplementary Materials.

Minor Comment 1

Lines 20, 21, 99, 100: What do the scales of the ULF and EMIC waves (~ 105 and 103 km) mean? Since the wavelength of the EMIC wave was estimated as 2700 km, I think that they might mean the wavelength. If so, is it possible to say anything about the wavelength of the ULF wave from the observations? For example, Kitamura et al. (2021, JGR) estimated a wavelength of ULF waves as ~ 3600 – 9000 km using the data obtained by the MMS spacecraft, although the wavelength is much smaller than 105 km. If the wavelength was very long, no difference must be visible between the spacecraft. If so, I think that you can write that. Since the scales are important in this manuscript, it is good to define and demonstrate the scales as carefully as possible.

Thanks for identifying this issue. In this study, the wavelength is used as a proxy for the spatial scales of the waves. Also, no difference is visible between the four MMS spacecraft (both for the ULF waves and the EMIC waves).

Here, the wavelength of the ULF waves is estimated from the dispersion relation for Alfvén waves, $\omega = v_A k_\parallel$. In this event, the Alfvén velocity v_A is ~ 662 km/s, as the amplitude of the background magnetic field is ~ 45 nT and the plasma number density is ~ 2.2 cm $^{-3}$. On the

other hand, the angular frequency of the ULF waves is about $\frac{2\pi}{6 \text{ min}} = 0.017 \text{ rad/s}$. Thus, the wavelength of the ULF waves is $\frac{2\pi}{k_{\parallel}} = 2.4 \times 10^5 \text{ km}$. This is why we say that the scale of the ULF waves is on the order of 10^5 km .

We have added a new subsection titled “Estimating the spatial scales of waves” in the Supplementary Materials. The technical definition of the spatial scale of waves, together with how it is obtained, is given in this subsection. Please find the details in lines 156-164 in the revised Supplementary Materials.

Minor Comment 2

Line 50: Since this is a magnetospheric phenomenon, to show the spacecraft location in the magnetic latitude, magnetic local time, and L-value coordinate in addition to that in the GSE coordinates is useful for readers. Since it is a bit far from the Earth, I think that it is good to also add information about the distance and relative location (north or south) to minimum-B at low latitudes. (The location of minimum-B in the MMS MEC data may be usable.)

Thanks. These parameters are now given in lines 103-106.

Minor Comment 3

Lines 92-93: I suggest adding the wave frequency normalized by the cyclotron frequency of H⁺.

Have done. Please find it in line 152.

Minor Comment 4

Lines 197-198: If my understanding is correct, “besides lower ~” is not come from the present result. To avoid misunderstandings, it would be better to emphasize more that it comes from other results.

Have done. Please find lines 267-268 in the manuscript.

Minor Comment 5

Lines 216-217: What do the curves in Fig. 2b indicate?

The two curves are just used to guide the eyes. They do not indicate anything specially. We have added a brief description in line 316.

Minor Comment 6

Figures, Figures in Supplementary Material: There is no clear statement about the source of the data (one of the spacecraft or average of all spacecraft?). It is also better to describe the observation mode (fast survey or burst).

Thanks very much for the suggestions. In this study, data (both for fields and particles)

averaged across all four MMS spacecraft is used. We have added some descriptions of the data source and the observation mode to the manuscript. Please find them in figure captions and lines 165-170 in the Supplementary Materials.

Minor Comment 7

Supplementary Material, Line 26: For the calculation of He⁺ anisotropy, what was used for the calculation as $f_{||}$? Any description is needed.

For all the panels shown in Supplementary Fig. 1, the anisotropy is defined as $(f_{\perp} - f_{||})/(f_{\perp} + f_{||})$, where $f_{||}$ and f_{\perp} represent the PSDs for field-aligned motion (0°-30° and 150°-180° PA) and perpendicular motion (75°-105° PA), respectively. The manuscript has been revised accordingly. Please find line 23 in the Supplementary Materials.

Minor Comment 8

Supplementary Material, Fig. 5: What does the color scale mean? (What 0% and 100% mean?)

The color code represents the normalized PSDs, which are defined as $\frac{f(E,\alpha,\phi_i)}{\sum_i f(E,\alpha,\phi_i)}$, where f denotes PSDs, E and α represent energy and pitch angle (fixed in the normalization processes), respectively, and ϕ_i represents the i th gyrophase channel. As defined in this way, 0% means that the PSDs in this gyrophase channel are zero, and 100% means that the PSDs in other gyrophase channels are zero. We have added a brief description of this point in the manuscript. Please find lines 321-322 in the main manuscript and lines 212-213 in the Supplementary Materials.

Minor Comment 9

Supplementary Material, Lines 120-128: It is necessary to explain why you can ignore He⁺ (and O⁺).

Thanks. According to the HPCA observations, the total number density of He⁺ and O⁺ ions are only ~2% and ~1% of that of H⁺ ions, respectively. Thus, to a first order approximation, the heavy ions would not affect the dispersion relation much and thus can be ignored. We have added a brief discussion of this point in lines 149-151 in the Supplementary Materials.

Minor Comment 10

Supplementary Material, Lines 120-128: To describe the minimum resonant energy for H⁺ is useful for Supplementary Material Fig. 1e.

Have done. Please find lines 33-34 in the Supplementary Materials.

Minor Comment 11

Supplementary Material, Lines 139-143: The description about FPI is incorrect. The energy range does not agree with Fig. 1a (2 eV?–20 keV?). The energy range does not appear to be log-evenly divided at least below 10 eV (Fig. 1b). The use of the two energy tables for the FPI burst data was

supposed to have been stopped in 2016. Some description about the fast survey data is also necessary, if they are also used.

Sorry for the mistakes. We have corrected the mistakes and added a brief description of the fast survey data in lines 183-184 in the Supplementary Materials.

Minor Comment 12

Supplementary Material, Lines 169-181: The energy and angular resolution of HPCA data (fast survey only?) should be described.

Have done. Please find the descriptions in lines 220-221 in the Supplementary Materials.

Correction

1. Supplementary Material, Line 24: 30° - 30° -> 30° - 150° ?

It should be 75° - 105° pitch angles. Please find line 25 in the Supplementary Materials.

2. Supplementary Material, Line 40: Missing description about panel (b)

The description has been added. Please find line 40 in the Supplementary Materials.

3. Supplementary Material, Line 137: 15 -> 150

Have been corrected.

4. Supplementary Material, Line 174: HPAC -> HPCA

Have been corrected.

Reviewers' comments:

Reviewer #1 (Remarks to the Author):

I have read the responses the authors have made and I feel that on balance the paper is essentially ready for publication from my point of view. I feel this study is an important confirmation of the energy cascade processes, despite the fact that some aspects have been reported in other work (but not with the depth of analysis given here). They have made corrections along the lines I suggested (even though I indicated these were optional).

The most important addition is to include a second event which shows that similar processes operate (given the context of slightly different event conditions). The authors also reasonably point out the limitations of the datasets currently available so that other events often do not have the data resolution needed for a full analysis.

The addition of another event shows that the exact nature of the cross-scale transfer depends on input conditions to some extent and I feel this also addresses the criticism that other work has dealt with this topic. In this work, common mechanisms (as well as differences) are shown through an event comparison, as well as utilising high-precision, multi-spacecraft data to give direct evidence, which was not carried out before.

I recommend publication.

Reviewer #2 (Remarks to the Author):

The authors have addressed my previous comments and supported their conclusions by extended analysis. However, I still believe this manuscript would be more suitable for a more specialized journal, for example, Geophysical Research Letters or Journal of Geophysical Research as it does not represent "important advances of significance to specialists in the field" to justify publication in Nature Communications.

Reviewer #3 (Remarks to the Author):

Regrettably, based on the newly added sections, I have come to suspect that it may be difficult to clearly demonstrate what you are trying to prove in the present manuscript, although I understand the importance of the topic.

Lines are for the text with track changes.

Major Comments

Comment 1

Lines 59-100: In this section, you attempt to argue that H^+ ions are accelerated by the ULF waves. However, H^+ ions, inversely, may excite the ULF wave. Tian et al. (2022) and Kitamura et al. (2021) indicate that (mainly) compressional ULF waves may be generated internally by anisotropic ions. If their expectation is correct, the direction of energy transfer is opposite to what is argued in the present manuscript. The situation seems to be similar to the events studied by them, since there appears to be a significant compressional component in all events, except for the inappropriate events (2 and 5 in Table 1). A good correlation between the ULF wave and EMIC waves does not indicate the direction of energy transport. In the manuscripts by Min et al. (2017), Takahashi et al. (2018), and Yamamoto et al. (2019) which are referred to in the present manuscript, resonant ions have been discussed as a possible energy source for the ULF waves. If the ULF waves are apparently externally excited (Zong et al., 2012; Wang et al., 2021), energy transport is probably directed from the ULF waves to ions. Since the events are quiet condition, this would not be the case without any evidence. Thus, the direction of energy transport is at least not obvious. I think that it is impossible to conclude that the ions were accelerated by the ULF waves unless you rule out internal excitation of the ULF wave by ions or show any evidence of energy transport from the ULF waves to the ions. Although I understand that this is a rather difficult requirement, since the main conclusion is the transport of energy from macroscales to microscales, any possibility or idea that contradicts the scenario must be eliminated.

Tian et al. (2022). Structure of Pc 5 compressional waves observed in the duskside outer magnetosphere: MMS observations. *Journal of Geophysical Research: Space Physics*, 127, e2021JA029817. <https://doi.org/10.1029/2021JA029817>

Kitamura et al. (2021). Energy transfer between hot protons and electromagnetic ion cyclotron waves in compressional Pc5 ultra-low frequency waves. *Journal of Geophysical Research: Space Physics*, 126, e2020JA028912. <https://doi.org/10.1029/2020JA028912>

Comment 2

Lines 142-143: If the azimuthal wavenumber was $\sim 80\text{--}7000$ as expected, the azimuthal wavelength became $\sim 75\text{--}6500$ km around the radial distance of $\sim 13 R_E$. If the wavelength was short, a slight phase difference in the ULF wave among spacecraft would be visible, which is inconsistent with the observed fact that no difference is seen among the spacecraft. If the wavelength was really so short, I wonder if the finite Larmor radius effect can be used to estimate the wavelength (Min et al., 2017; Takahashi et al., 2018; Yamamoto et al., 2019; Kitamura et al., 2021).

Comment 3

Lines 176-181: Although the method suggested by Blum et al. (2009) is used to show the instability quantitatively, there are concerns on two points. The first point is that the ion species are different. Since the original formula was made for H^+ ions, it is not obvious whether the same formulas, especially for S and α , can be used for He^+ . At least, justification and a more detailed description of the formulas, including coefficients for S and α , are needed. The other point is that the formula was made for a single ion species. It is also questionable whether this formula can be used directly for minor He^+ ions (with dominant H^+ ions).

Minor comments

Comment 1

Lines 63 and 105: Since it is a bit far from the Earth, I think that it is good to also add information about the distance and relative location (north or south) to minimum- B . (In the outer magnetosphere, 0° in the magnetic latitude in the MMS MEC data does not indicate the location of minimum- B . The location of minimum- B is provided separately in the MEC data.)

Comment 2

Line 316 and Supplementary materials Line 67: I think that the HPCA data used for those figures are in the fast survey mode.

Comment 3

When calculating the rest frame of the plasma for FPI data, it is necessary to make an assumption about ion species. Is it assumed that they are all He^+ ? Furthermore, since EMIC waves cause electric field fluctuation with a short period, it is necessary to be careful in handling the electric field data to determine the rest frame of the plasma. I think that it is good to describe how the rest frame of the plasma was determined for each of the ion measurements with various time resolutions.

Comment 4

Fig. 4, Supplementary materials Fig. 6, and Supplementary materials Lines 211-212: Since the number of bins in the gyrophase direction is 12, I thought that the average should become $\sim 8.3\%$, which is out of the range of the color bar of the figures. Are they correct?

Comment 5

Table 1 (Line 78): Although Event 2 has burst data, the increase in wave intensity in the frequency range of EMIC waves seems to be caused by contacts with the boundary layer due to the ULF wave. I consider that this event is unsuitable for the present analysis.

Comment 6

Table 1: The date of Event 5 is wrong.

Comment 7

Supplementary Materials Lines 218-220: The angular resolution of the fast survey data is 45° in one of the directions due to the data decimation. Energy bins are also reduced due to the data decimation. I think that it is good to follow the description in the Data Product Guide for HPCA.

https://lasp.colorado.edu/mms/sdc/public/datasets/hpca/10160.13-MMS-HPCA_SCI_ALG_UM_20160310_0.pdf

Reviewer #1 (Remarks to the Author):

I have read the responses the authors have made and I feel that on balance the paper is essentially ready for publication from my point of view. I feel this study is an important confirmation of the energy cascade processes, despite the fact that some aspects have been reported in other work (but not with the depth of analysis given here). They have made corrections along the lines I suggested (even though I indicated these were optional).

The most important addition is to include a second event which shows that similar processes operate (given the context of slightly different event conditions). The authors also reasonably point out the limitations of the datasets currently available so that other events often do not have the data resolution needed for a full analysis.

The addition of another event shows that the exact nature of the cross-scale transfer depends on input conditions to some extent and I feel this also addresses the criticism that other work has dealt with this topic. In this work, common mechanisms (as well as differences) are shown through an event comparison, as well as utilizing high-precision, multi-spacecraft data to give direct evidence, which was not carried out before.

I recommend publication.

We are very grateful to reviewer #1 for his/her continued efforts in evaluating this paper.

Reviewer #2 (Remarks to the Author):

The authors have addressed my previous comments and supported their conclusions by extended analysis. However, I still believe this manuscript would be more suitable for a more specialized journal, for example, Geophysical Research Letters or Journal of Geophysical Research as it does not represent “important advances of significance to specialists in the field” to justify publication in Nature Communications.

We thank Reviewer #2 for his/her continued efforts in reviewing this manuscript. However, we strongly disagree with the judgment presented here. We are emphatic that our manuscript is entirely different from the two previous publications (Kitamura et al. [2018; DOI: 10.1126/science.aap8730] and Shi et al. [2020; DOI: 10.1063/1.5142686]) for reasons given below.

The question we investigate is that, besides turbulent cascade, could any other processes mediate cross-scale energy transfer in collisionless plasmas? With direct in-situ measurements, we demonstrate for the first time that the cross-scale interactions between hot ions, macroscale ULF waves, and microscale EMIC waves are such a candidate, since direct energy transfer from macroscales down to microscales can take place via this mechanism. This newly identified “cross-scale wave-particle interaction” mechanism provides a novel basis for understanding cross-scale energy transfer processes in space and astrophysical plasma systems. In contrast, Kitamura et al. only concentrate on a microscale process, the EMIC wave-ion cyclotron resonance. They neither studied macroscale processes nor cross-scale processes. Also, we focus on very different physics from Shi et al., although we both analyzed the same event. Shi et al. studied the behavior of cold ions in EMIC wave fields. They neither discussed nor identified the cross-scale coupling processes controlling the event. Therefore, we are emphatic that our manuscript is entirely different from these two papers.

Reviewer #3 (Remarks to the Author):

Regrettably, based on the newly added sections, I have come to suspect that it may be difficult to clearly demonstrate what you are trying to prove in the present manuscript, although I understand the importance of the topic.

We are very grateful to the reviewer for the constructive comments and the judgment of the importance of our manuscript's topic. We have revised the manuscript according to the comments listed here. Especially, we have calculated the local energy flow between the ULF waves and ions, using the observed electric fields and ion velocity distributions. The results suggest that the energy transport is indeed directed from ULF waves to ions, when EMIC waves are observed. The new results well support our two-step mechanism: The ULF waves first quasi-periodically accelerate hot ions and increase their anisotropy, and then, the anisotropic ions quasi-periodically excite/amplify the EMIC waves. Please see the detailed response below.

(Lines are for the text with track changes.)

Major Comments

Comment 1

Lines 59-100: In this section, you attempt to argue that H^+ ions are accelerated by the ULF waves. However, H^+ ions, inversely, may excite the ULF wave. Tian et al. (2022) and Kitamura et al. (2021) indicate that (mainly) compressional ULF waves may be generated internally by anisotropic ions. If their expectation is correct, the direction of energy transfer is opposite to what is argued in the present manuscript. The situation seems to be similar to the events studied by them, since there appears to be a significant compressional component in all events, except for the inappropriate events (2 and 5 in Table 1). A good correlation between the ULF wave and EMIC waves does not indicate the direction of energy transport. In the manuscripts by Min et al. (2017), Takahashi et al. (2018), and Yamamoto et al. (2019) which are referred to in the present manuscript, resonant ions have been discussed as a possible energy source for the ULF waves. If the ULF waves are apparently externally excited (Zong et al., 2012; Wang et al., 2021), energy transport is probably directed from the ULF waves to ions. Since the events are quiet condition, this would not be the case without any evidence. Thus, the direction of energy transport is at least not obvious. I think that it is impossible to conclude that the ions were accelerated by the ULF waves unless you rule

out internal excitation of the ULF wave by ions or show any evidence of energy transport from the ULF waves to the ions. Although I understand that this is a rather difficult requirement, since the main conclusion is the transport of energy from macroscales to microscales, any possibility or idea that contradicts the scenario must be eliminated.

We sincerely appreciate the comments. We totally agree that we cannot identify unambiguously the source of the ULF waves here, because of the lack of necessary information (e.g., radial gradient of ion phase space densities). However, respectfully, we disagree with the suggestion that “the direction of energy transfer is opposite to what is argued in the present manuscript”, if the ULF waves are generated internally. In the revised manuscript, we have calculated the local energy flow ($eE_w \cdot v_i$) between the ULF waves and ions, using the in-situ measurements of ULF wave electric fields and ion velocity distributions (from FPI). Figure 1f shows the obtained energy gain per ion per unit time for ions in the energy range of 9.8-26.6 keV. We can see that, when the EMIC waves are observed, the energy flow is directed from the ULF waves to ions (i.e., the energy gain is positive). Thus, the new observations support our scenario.

We note that the mechanism proposed in our manuscript actually does not require any specific sources of ULF waves. It can also happen when ULF waves are internally excited. In this situation, ULF waves are still able to periodically transfer energy to H⁺ ions, increase ion anisotropy and then cause local EMIC wave growth. During the growth of the EMIC waves, part energy irreversibly flows from ions to the EMIC waves, and does not return to the ULF waves anymore. In this way, energy transfer proceeds from macroscales to microscales. Our explanations still match the observations in this situation. The key point here is that the energy exchange between EMIC waves and ions is more localized (both spatially and temporally) than that between the ULF waves and ions. Although the ultimate energy source might be ions, ULF waves can still transfer energy to ions and make the latter act as an energy source of EMIC waves at a given time and location. Nevertheless, regarding our event, we suggest that the actual source of the ULF waves is not an essential issue, since the direct observations show that the energy is flowing from ULF waves to ions, when EMIC waves are observed.

We have added some new discussions in the revised manuscript accordingly. Please find lines 71-74:

...With the PSD oscillations, H⁺ ions exchange energy with the ULF waves, as shown in Fig.

1f which gives ion local energy gain ($eE_w \cdot v_i$) calculated from the observed ULF-wave electric fields (E_w) and ion velocity (v_i) distributions. Particularly, we note H⁺ ions gain energy from ULF waves when their PSDs peak....

and lines 99-105:

It is noted that the mechanism suggested here does not require any specific sources of the ULF waves. Especially, it can happen even when ULF waves are internally excited by ions¹¹⁻¹⁴, since in this situation, ULF waves are still able to periodically transfer energy to H⁺ ions and increase their anisotropy, making them excite EMIC waves. During the growth of EMIC waves, part energy flows to EMIC waves and does not return to ULF waves anymore. In this way, energy transfer proceeds from macroscales to microscales, although in this case, the ultimate energy source is ions and ULF waves only act as an intermediate step.

Comment 2

Lines 142-143: If the azimuthal wavenumber was $\sim 80-7000$ as expected, the azimuthal wavelength became $\sim 75-6500$ km around the radial distance of $\sim 13 R_E$. If the wavelength was short, a slight phase difference in the ULF wave among spacecraft would be visible, which is inconsistent with the observed fact that no difference is seen among the spacecraft. If the wavelength was really so short, I wonder if the finite Larmor radius effect can be used to estimate the wavelength (Min et al., 2017; Takahashi et al., 2018; Yamamoto et al., 2019; Kitamura et al., 2021).

Thanks very much for identifying this issue. Indeed, as suggested by the reviewer, signatures like phase shift should be observed, if the wavelength of the ULF waves was really so short. However, we have rechecked the data and did not find any such signals. Therefore, in the revised manuscript, we choose to use a more general term, “drift-bounce interactions”, instead of the old one, “drift-bounce resonance”.

However, we suggest that our conclusions are not affected much. Now, we have calculated the energy flow between the ULF waves and He⁺ ions, using the directly measured electric fields and ion velocity distributions (HPCA fast survey data). As shown in Supplementary Fig. 1c, the results suggest the energy is directed from the ULF waves to He⁺ ions when the three EMIC wave packets are observed. Thus, the new direct measurements support our conclusions that the ULF waves are transferring energy to He⁺ ions, when EMIC waves are observed.

In addition, we suggest that the scenario proposed in our manuscript actually does not require resonant interactions between ULF waves and ions. Without resonance, ULF waves can also accelerate ions in the perpendicular direction, increase their anisotropy to exceed the EMIC wave-instability threshold, and then lead to EMIC wave growth. (Of course, without resonance and EMIC wave excitation, the energy from ULF waves to ions will fully return to ULF waves in the next half wave cycle.) Thus, our scenario can occur without resonance. Finally, we would like to highlight here that our scenario provides a simple, straightforward but complete explanation for all the various observations: the quasi-periodic perpendicular acceleration of He⁺ ions by ULF waves (Fig. 2c and Supplementary Fig. 1c), the quasi-periodic increase of He⁺ ion anisotropy (Supplementary Fig. 1d and 1e), the EMIC wave packets appearing in coincidence with the ULF wave fields (Figure 2d and Supplementary 2d), the signatures indicative of EMIC wave-ion cyclotron-resonance (Fig. 4), and the secular energy flow directed from ions modulated by ULF waves to EMIC waves (Fig. 4). The results reported here present some observational evidence for the capability of cross-scale wave-particle interactions of mediating cross-scale energy transfer, and would advance our understanding of relevant processes in space plasma systems.

We have revised the manuscript accordingly. Please find lines 146-156:

Hence, instead of $E \times B$ drift, the observed PSD enhancements should be caused by another type of ULF wave-particle interactions—drift-bounce interactions^{16,20,21}. This type of interactions occur between ULF waves and particles' drift and bounce motion, and would preferentially accelerate particles in the perpendicular direction. The efficiency of the interactions peaks when they take the form of resonance. However, because of the complex configuration of the dayside outer magnetosphere (e.g., the existence of off-equatorial magnetic minima²²), we cannot either confirm or rule out unambiguously the occurrence of resonance in this event. Nevertheless, this does not affect our analysis much, since what is essential here—local energy flow from ULF waves to He⁺ ions and ULF wave-induced enhancements in He⁺-ion anisotropy—has been directly observed (shown below).

Comment 3

Lines 176-181: Although the method suggested by Blum et al. (2009) is used to show the instability quantitatively, there are concerns on two points. The first point is that the ion species are different. Since the original formula was made for H⁺ ions, it is not obvious whether the same

formulas, especially for S and α , can be used for He^+ . At least, justification and a more detailed description of the formulas, including coefficients for S and α , are needed. The other point is that the formula was made for a single ion species. It is also questionable whether this formula can be used directly for minor He^+ ions (with dominant H^+ ions).

We sincerely appreciate this comment. We agree that applying the method suggested by Blum et al. (2009) here is not appropriate. Actually, after careful reconsideration, we think it is better to remove the relevant text, and instead only give a more general, qualitative discussion about the instabilities. Our consideration is primarily based on the following reasons.

The first reason is that it is not appropriate to use any linear or quasi-linear instability formula here, since the EMIC wave-ion cyclotron interactions in this event are very nonlinear. As an illustration, we can calculate the energy change (δW) of a 1 keV (W) ion over a gyro-period. Given a wave electric field of $E_w \approx 1$ mV/m (Supplementary Figure 4f) and a background magnetic field of $B_0 \approx 47$ nT, δW is estimated to be $2\pi \frac{mv}{eB_0} \cdot eE_w = 2\pi \frac{E_w}{B_0} \sqrt{2mW} \approx 1.2$ keV, which is comparable with the initial energy W . (Please note that the energy gain shown in Figure 4b has been averaged over gyro-phases, and thus should be much less than the energy change of an individual ion.) Thus, the interactions between the waves and ions are very nonlinear, and cannot be treated linearly or quasi-linearly. This suggestion is supported by the observations of gyro-phase bunching, which indicate the interactions are indeed very nonlinear. As shown in previous theoretical studies and numerical simulations [e.g., Albert 2002, Bortnik et al. 2008, and references therein], wave-particle energy exchange in nonlinear interactions is very different from that in linear/quasi-linear interactions (Generally, energy exchange in the former is much larger than that in the latter). Hence, it is not appropriate to apply any linear or quasi-linear instability analyses here. On the other hand, at present, nonlinear instabilities generally can only be quantitatively analyzed via hybrid or PIC simulations, which are much beyond our scope here.

The second reason is that there is no need to seek help from quantitative instability analysis here. As shown in the manuscript, we have already directly observed the energy exchange between the EMIC waves and particles, and find the observed energy exchange is large enough to cover the growth of the EMIC waves (Fig. 4). Thus, unlike in previous studies where direct observations are unavailable, here we can conclude from observations themselves that the hot anisotropic He^+ ions observed can provide enough free energy for the growth of the EMIC

waves.

Therefore, in accordance with the above argument, we decided to remove the text related to quantitative instability analysis from the manuscript. However, we still give a detailed qualitative discussion about the instabilities, especially with respect to the shape of ion velocity distribution functions. The relevant discussion can be found in lines 186-193 of the revised manuscript and below:

As shown in what follows, gyrophase bunching indicative of strong nonlinearity is observed in this event. Hence, it is not appropriate to apply any linear or quasi-linear instability analyses (e.g., ref.²⁴) here. To reasonably describe the nonlinear ICA instability, hybrid or particle-in-cell numerical simulations are needed in general, which are much beyond our scope here. On the other hand, the local energy flow for EMIC wave growth is observed here directly and is sufficiently large. Thus, unlike in previous studies without direct observations, here we can conclude from observations themselves that the hot anisotropic He⁺ ions observed can provide enough free energy for the growth of the EMIC waves.

Reference:

Albert, J. M. (2002). Nonlinear interaction of outer zone electrons with VLF waves. Geophysical research letters, 29(8), 116-1.

Bortnik, J., Thorne, R. M., & Inan, U. S. (2008). Nonlinear interaction of energetic electrons with large amplitude chorus. Geophysical Research Letters, 35(21).

Minor comments

Comment 1

Lines 63 and 105: Since it is a bit far from the Earth, I think that it is good to also add information about the distance and relative location (north or south) to minimum-*B*. (In the outer magnetosphere, 0° in the magnetic latitude in the MMS MEC data does not indicate the location of minimum-*B*. The location of minimum-*B* is provided separately in the MEC data.)

Thanks very much. The relative locations of the spacecraft to minimum-*B* (whose locations are read from the MEC data) have been given in the revised manuscript. In the first event observed on September 5, 2015, the spacecraft was located ~10.5 degrees south of minimum-*B*, whereas

in the second event observed on January 7, 2019, the spacecraft was located ~ 9.8 degrees south of minimum-B.

Please find lines 62-65 in the revised manuscript:

This event was observed in the duskside magnetosphere (GSE [4.1, 10.2, 0.0] Earth radius, L-shell ~ 12.2 and magnetic local time ~ 18.0 hr) on September 5, 2015. At this time, MMS was located $\sim 10.5^\circ$ south of the magnetic equator (taken as the minimum-B point given in the MMS/MEC data).

and also lines 113-115:

In this event, MMS was located in the outer duskside magnetosphere (GSE [6.2, 7.8, 1.0] Earth radius, L-shell ~ 10.3 and magnetic local time ~ 15.1 hr), and $\sim 9.8^\circ$ south of the magnetic equator.

Comment 2

Line 316 and Supplementary materials Line 67: I think that the HPCA data used for those figures are in the fast survey mode.

Thanks very much for identifying this issue, and sincerely sorry for the mistakes. Fast survey mode data is used for these figures. We have revised the manuscript accordingly. Please find lines 323-324 in the main manuscript and lines 66-67 in the Supplementary Materials.

Comment 3

When calculating the rest frame of the plasma for FPI data, it is necessary to make an assumption about ion species. Is it assumed that they are all He⁺? Furthermore, since EMIC waves cause electric field fluctuation with a short period, it is necessary to be careful in handling the electric field data to determine the rest frame of the plasma. I think that it is good to describe how the rest frame of the plasma was determined for each of the ion measurements with various time resolutions.

We are grateful for the suggestions. The rest frame of the plasma is used for three figures: Supplementary Fig. 1d-1f, Supplementary Fig. 3 and Fig. 4b. For all the three uses, the frame is determined from direct plasma measurements, rather than electric field data. Now, the working definitions of these frames are given in lines 239-257 in the Supplementary Materials:

The rest frame of the plasma is used when calculating the anisotropy of H⁺ and He⁺ ions (Supplementary Fig. 1 and 3), and the energy exchange between He⁺ ions and EMIC waves

(Fig. 4b). The working definitions of these frames are:

Supplementary Fig. 1. For Supplementary Fig. 1c presenting He⁺ ion anisotropy, the rest frame of the plasma is determined according to the bulk velocity of He⁺ ions given in the HPCA fast survey mode data. For Supplementary Fig. 1e showing H⁺ ions, the rest frame of the plasma is determined according to the bulk velocity of H⁺ ions given in the HPCA fast survey mode data.

Supplementary Fig. 3. Here, the rest frame of the plasma is defined according to the He⁺ ion bulk velocity given in the HPCA fast survey mode data. The frame transformation is applied prior to the time average.

Fig. 4b. Here, the rest frame of the plasma is determined from the He⁺ ion bulk velocity given in the HPCA fast survey mode data. We first applied a low-pass filter with an upper cutoff frequency of 0.05 Hz (six times less than the frequency of the EMIC waves) to the bulk velocity during 20:39:40-20:40:40 UT. Then, the resulting bulk velocity was linearly interpolated onto the epoch series of FPI burst mode data. The obtained bulk velocity series is then used to calculate the rest frame of the plasma. (As determined in this way, there is no ULF electric field in this frame of reference.)

Comment 4

Fig. 4, Supplementary materials Fig. 6, and Supplementary materials Lines 211-212: Since the number of bins in the gyrophase direction is 12, I thought that the average should become ~8.3%, which is out of the range of the color bar of the figures. Are they correct?

We sincerely appreciate the careful review, and are sorry for the mistakes. The color bars shown in the old version of the manuscript are incorrect. These wrong color bars are generated for figures in which the gyrophase is divided into ten bins (this is why they are centered at ~10%). When we updated figures, we forgot to update color bars simultaneously. We have corrected the mistakes in the revised manuscript. Also, since here only the relative values are of importance, the mistakes do not affect our analysis and conclusions.

Comment 5

Table 1 (Line 78): Although Event 2 has burst data, the increase in wave intensity in the frequency

range of EMIC waves seems to be caused by contacts with the boundary layer due to the ULF wave. I consider that this event is unsuitable for the present analysis.

We agree with the reviewer. We have removed this event from the table.

Comment 6

Table 1: The date of Event 5 is wrong.

Sincerely sorry for this mistake. This wrong event has been removed from the table. We have rechecked the remaining events and the parameters.

Comment 7

Supplementary Materials Lines 218-220: The angular resolution of the fast survey data is 45° in one of the directions due to the data decimation. Energy bins are also reduced due to the data decimation. I think that it is good to follow the description in the Data Product Guide for HPCA. https://lasp.colorado.edu/mms/sdc/public/datasets/hpca/10160.13-MMS-HPCA_SCI_ALG_UM_20160310_0.pdf

Thanks very much for the comments. The old version of the descriptions is based on CDF files. We have updated the relevant text following the Data Product Guide for HPCA. Please find lines 217-220 in the Supplementary Materials.

Thus, only fast survey mode data is used in this paper. In this mode, the HPCA instruments provide a 4π measurement of ions from ~ 1.4 eV/q to 37 keV/q every 10 seconds (i.e., the 1/2 spin of the spacecraft). Taking into account the data decimation, the effective angular resolution of the data is about 45° , and the effective number of the energy channels is 16.

REVIEWER COMMENTS

Reviewer #3 (Remarks to the Author):

I understand the importance of the topic, and I think that the revision is in the right direction, but the lack of important information in the text makes it impossible to evaluate whether the results are valid or not in the current situation.

Please see my comments below.

I understand the importance of the topic, and I think that the revision is in the right direction, but the lack of important information in the text makes it impossible to evaluate whether the results are valid or not in the current situation.

Lines are for the text with track changes.

Major Comments

Comment 1

Line 73: The definition of the observed ULF-wave electric field (E_w) is not written. Is the component in any frequency range extracted? Excluding the contribution of DC component is important to exclude the effect of non-zero (instrumental or physical) offset.

$$E = E_{DC} + E_{wave}$$

$$v_i = v_{i_DC} + v_{i_wave}$$

Even if the dot product of E_{DC} and v_{i_DC} is zero, the dot product of E and v_i is not equal to E_{wave} and v_{i_wave} (energy transfer between ULF wave components). There may be contributions from the dot products of E_{wave} and v_{i_DC} or E_{DC} and v_{i_wave} .

Comment 2

Lines 100-102: As discussed in my previous comments, I think that one should not ignore the possibility that the ULF wave is primarily a compressional mode structure. If the ULF wave is mainly compressional mirror-mode like structures (almost zero frequency in plasma rest frame), it seems more correct to say that the energy is transferred gradually as the decay of the structures rather than periodically. If it is not the case, it is only necessary to show that the compressional component was not dominant. I think that it will be more reader-friendly and the physics easier to understand if the electromagnetic field is converted to the field-aligned coordinate system (toroidal, poloidal, and compressional components) in the entire discussion related to the ULF waves. Instead of Fig. 1a, I suggest plotting the three components of the magnetic field variation in the field-aligned coordinates and the three components of the electric field variation ($=E_w$?) in the field-aligned coordinates with two panels. If there is a possibility that the compressional component was dominant, it seems appropriate to cite a couple of papers on compressional ULF waves (structures) in addition to those on the other modes that have already been cited.

Comment 3

I think that it will help to understand the variation of P if you show not only the electric field (E_w) but also three components of v_i in the field-aligned coordinates in Fig. 1. As far

as I know, there is no previous study on energy transport related to ULF waves. Thus, I suggest showing and briefly discussing how (positive) P is generated from E_w and v_i , for example, by showing the phase relationship between E_w and v_i . If you do not discuss them, you should demonstrate the significance of P (magnitude and/or phase difference between E_w and v_i), although I understand that P does not depend on specific physical process.

Comment 4

Is $\delta E_{ULF \rightarrow i}$ different from P? Although it is a critical parameter, there is no description of how it is calculated. Since there is no way to know which time intervals were used to derive the values, it is unlikely that the reader will be able to reproduce it. The significance of those values is also unclear. Since the parameters shown in the table are complex and the number of events has been reduced to only three, I think that it might be easy to understand for readers to show three additional supplementary figures in the same format (after update) as Fig. 1. If the table is kept as it is, I think it will be better to replace the components written in the ULF Field with those in the field-aligned coordinates (toroidal, poloidal, or compressional), and for $\delta E_{ULF \rightarrow i}$, the time interval used for the calculation should be indicated in each added figure with the description in the figure caption. It may be worthwhile to add the events treated in the main text to the table.

Minor comments

Comment 5

Since it has changed to treat not only EMIC waves but also ULF waves in detail in this revision, the use of E_w , E_{wave} , B_{wave} , and f_{wave} , etc. makes it difficult for readers to understand which wave they are related to. I suggest replacing them with descriptions (e.g., E_w_{EMIC} or E_w_{ULF}) that makes it easy to identify immediately which wave they are related to.

Comment 6

Using E for both electric field and energy is confusing. The same is true for the use of f for PSD and frequency. The use of P for energy transfer is also slightly confusing with Power. I recommend improving on such representations. (I apologize for oversight in pointing this out in my previous review.)

Response to Reviewer #3:

We are very grateful to reviewer #3 for his/her continued efforts in evaluating this paper. We also sincerely appreciate his/her constructive comments, which greatly helped us improve the manuscript. Now, we have carefully considered these comments and revised the manuscript accordingly. Please find details in the following letter and the revised manuscript. (Please note that line numbers are for the text with track changes.)

I understand the importance of the topic, and I think that the revision is in the right direction, but the lack of important information in the text makes it impossible to evaluate whether the results are valid or not in the current situation. (Lines are for the text with track changes.)

All materials suggested by the reviewer have been added to the manuscript.

Major Comments

Comment 1

Line 73: The definition of the observed ULF-wave electric field (E_w) is not written. Is the component in any frequency range extracted? Excluding the contribution of DC component is important to exclude the effect of non-zero (instrumental or physical) offset.

$$E = E_{DC} + E_{wave}$$
$$v_i = v_{i_DC} + v_{i_wave}$$

Even if the dot product of E_{DC} and v_{i_DC} is zero, the dot product of E and v_i is not equal to E_{wave} and v_{i_wave} (energy transfer between ULF wave components). There may be contributions from the dot products of E_{wave} and v_{i_DC} or E_{DC} and v_{i_wave} .

We agree with the reviewer and are sorry for the confusion. When calculating the ion energy gain ($eE_{w,ULF} \cdot v_{i,ULF}$), only components in the period range of 0.25-7 min were included (the lower and higher period limit are about twice the periods of the EMIC waves and the ULF waves, respectively). Both DC and EMIC-wave fields were excluded in the calculation, as now shown by the revised Fig. 1d and 1e.

We have revised the manuscript to show this point directly. Please find lines 70-71:

Here, a bandpass filter (0.25-7 min) is used when generating this panel.

and lines 315-316:

When generating panels c-f, a 0.25-7 min bandpass filter has been used.

Comment 2

Lines 100-102: As discussed in my previous comments, I think that one should not ignore the possibility that the ULF wave is primarily a compressional mode structure. If the ULF wave is mainly compressional mirror-mode like structures (almost zero frequency in plasma rest frame), it seems more correct to say that the energy is transferred gradually as the decay of the structures rather than periodically. If it is not the case, it is only necessary to show that the compressional component was not dominant. I think that it will be more reader-friendly and the physics easier to understand if the electromagnetic field is converted to the field-aligned coordinate system (toroidal, poloidal, and compressional components) in the entire discussion related to the ULF waves. Instead of Fig. 1a, I suggest plotting the three components of the magnetic field variation in the field-aligned coordinates and the three components of the electric field variation (=Ew?) in the field-aligned coordinates with two panels. If there is a possibility that the compressional component was dominant, it seems appropriate to cite a couple of papers on compressional ULF waves (structures) in addition to those on the other modes that have already been cited.

Thanks very much for the comments, and sorry that we did not catch this point in the previous review processes.

As suggested by the reviewer, we now plot in Fig. 1c the three components of the magnetic field variation (in the period range of 0.25-7 min) in a FAC system defined according to the local magnetic field averaged over 12:15-12:25 UT. (Please also see the ULF-wave electric fields in Fig. 1d, 1e, and Supplementary Fig. 1a.) One can see that Bp is indeed very significant in this event. It is even a little larger than the two transverse components. This observation suggests that the ULF waves could be a compressional mode structure. Thus, as suggested by the reviewer, we cite two papers on compressional ULF waves here (Kitamura et al. [2021] and Tian et al. [2022], which were mentioned in the previous review processes). Please find these revisions in lines 71-74:

We note that the parallel component of the ULF waves is very significant and even a little larger than the radial and azimuthal components, indicating the ULF waves are possibly compressional mode (e.g., mirror-mode structures^{7,8}).

However, we note that, for the event studied here, the amplitude of two transverse components of the magnetic field variation is comparable with that of the parallel component, unlike those reported in Kitamura et al. [2021] and Tian et al. [2022] in which the parallel component is definitely dominant. In addition, we also note that quasi-periodic energy flow from the ULF waves to nearby ions has been directly observed in this event, which, together with the anisotropy enhancements, are well correlated with the periodically appeared EMIC-wave packets. These measurements support our suggestions that the observed ULF waves are responsible for the periodical generation of the EMIC waves.

Comment 3

I think that it will help to understand the variation of P if you show not only the electric field (E_w) but also three components of v_i in the field-aligned coordinates in Fig. 1. As far as I know, there is no previous study on energy transport related to ULF waves. Thus, I suggest showing and briefly discussing how (positive) P is generated from E_w and v_i , for example, by showing the phase relationship between E_w and v_i . If you do not discuss them, you should demonstrate the significance of P (magnitude and/or phase difference between E_w and v_i), although I understand that P does not depend on specific physical process.

We sincerely appreciate the reviewer for the comments. The suggested materials have been added to the revised manuscript. The azimuthal and radial components of V_i can be found in Fig. 1d and 1e, together with the corresponding components of the electric fields. (The parallel component of V_i can be found in Supplementary Fig. 1b. Since the parallel electric field is almost zero, this component does not contribute to the ion energy gain.) As shown in Fig. 1e, V_r and E_r are highly correlated and almost in phase during the whole time interval of interest. This observation is supported by a cross-wavelet analysis of them, which, as shown in Supplementary Fig. 1c and 1d, gives a correlation coefficient of ~ 0.75 and phase shift of 24° at the period of the ULF waves (~ 3.4 min). As a result of the phase relationship, positive energy gain is generated. We note that the contribution from the radial component dominates the total energy gain. As shown in Supplementary Fig. 1e, the black curve, which represents the total energy gain, follows the red curve corresponding to the radial component well. On the other hand, V_a and E_a are not well correlated (Fig. 1d). The corresponding cross-wavelet correlation coefficient is only ~ 0.5 . Also, the contribution from the azimuthal component (the green curve in Supplementary Fig. 1e) to the total energy gain is generally smaller than that of

the radial component.

A discussion of this point has been added to the manuscript. Please find lines 97-104 in the revised manuscript

Fig. 1f shows the ion energy gain ($\frac{dW}{dt} = eE_{w,ULF} \cdot v_{i,ULF}$) calculated from the observed ULF-wave electric fields ($E_{w,ULF}$) and ion velocity ($v_{i,ULF}$) distributions. One can see that $\frac{dW}{dt}$ is positive when EMIC waves are significant. These positive values mainly result from the radial component of $E_{w,ULF}$ and $V_{w,ULF}$, which, as shown in Fig. 1e, are highly correlated and approximately in phase during the whole time period of interest (CC~0.76 and phase shift~24°; Supplementary Fig. 1c and 1d). On the other hand, the azimuthal component of $E_{w,ULF}$ and $V_{w,ULF}$ are not well correlated (CC<0.5). Also, their contribution to the $\frac{dW}{dt}$ is smaller than that of the radial component (Supplementary Fig. 1e).

Comment 4

Is $\delta E_{ULF \rightarrow i}$ different from P ? Although it is a critical parameter, there is no description of how it is calculated. Since there is no way to know which time intervals were used to derive the values, it is unlikely that the reader will be able to reproduce it. The significance of those values is also unclear. Since the parameters shown in the table are complex and the number of events has been reduced to only three, I think that it might be easy to understand for readers to show three additional supplementary figures in the same format (after update) as Fig. 1. If the table is kept as it is, I think it will be better to replace the components written in the ULF Field with those in the field-aligned coordinates (toroidal, poloidal, or compressional), and for $\delta E_{ULF \rightarrow i}$, the time interval used for the calculation should be indicated in each added figure with the description in the figure caption. It may be worthwhile to add the events treated in the main text to the table.

We are very grateful to the reviewer for the comments. $\delta E_{ULF \rightarrow i}$ denotes the average value of P over corresponding time intervals.

Please note that, to avoid any potential uncertainty and ambiguity, we have removed the three supplementary events from the Supplementary Materials in the new version of the manuscript. Our main concern is that, unlike the cases shown in the main text, the ULF waves in these supplementary events are not monochromatic, making it very hard to obtain the ion

energy gain related to the periodical EMIC-wave packets accurately. Therefore, we think it is better not to show these events. (We noted this issue when we were considering comment #1. We sincerely thank the reviewer for this comment.) Besides Supplementary Table 1, relevant discussion in the main text (e.g., “For example, we list another three similar events in Supplementary Table 1”, lines 124-126) has also been removed.

Minor comments

Comment 5

Since it has changed to treat not only EMIC waves but also ULF waves in detail in this revision, the use of E_w , E_{wave} , B_{wave} , and f_{wave} , etc. makes it difficult for readers to understand which wave they are related to. I suggest replacing them with descriptions (e.g., E_w _EMIC or E_w _ULF) that makes it easy to identify immediately which wave they are related to.

Thanks very much for the suggestion. We have revised the manuscript accordingly. Now, variables associated with ULF wave and EMIC wave are denoted with subscripts “ULF” and “EMIC”, respectively. Please see the manuscript for detail.

Comment 6

Using E for both electric field and energy is confusing. The same is true for the use of f for PSD and frequency. The use of P for energy transfer is also slightly confusing with Power. I recommend improving on such representations. (I apologize for oversight in pointing this out in my previous review.)

Thanks. All suggestions have been adopted. Now, “ E ”, “ W ”, “ f ” and “PSD” are used to represent the electric field, energy, frequency and PSDs. Also, instead of “ P ”, now the ion energy gain is denoted by “ dW/dt ”, from which one can easily get the meanings.

REVIEWERS' COMMENTS

Reviewer #1 (Remarks to the Author):

I feel that the current revision of this paper, together with its supplementary material, has clarified many of the points raised. They now focus on two key events and improve and clarify definitions of the quantities analysed. This adds significant weight to their conclusions. I therefore feel it is suitable for publication.

Reviewer #3 (Remarks to the Author):

I think that the authors addressed my suggestions adequately, and this manuscript can be accepted for publication in Nature Communications after a minor revision.

Minor comment

Figure 1: I think that a line plot of wave intensity variation is easier to understand for readers than a power spectrum. It would be good to recover the line plot showing wave intensity variation, which shows a clear correlation with dw/dt and A , that was previously removed.

Response to Reviewer #1:

I feel that the current revision of this paper, together with its supplementary material, has clarified many of the points raised. They now focus on two key events and improve and clarify definitions of the quantities analyzed. This adds significant weight to their conclusions. I therefore feel it is suitable for publication.

We are very grateful to reviewer #1 for his/her continued efforts in evaluating this paper.

Response to Reviewer #3:

I think that the authors addressed my suggestions adequately, and this manuscript can be accepted for publication in Nature Communications after a minor revision.

We are very grateful to reviewer #3 for his/her continued efforts in evaluating this paper. We have carefully considered the comments shown below and revised the manuscript accordingly.

Minor comment

Figure 1: I think that a line plot of wave intensity variation is easier to understand for readers than a power spectrum. It would be good to recover the line plot showing wave intensity variation, which shows a clear correlation with dw/dt and A , that was previously removed.

Thanks for the suggestion. We have recovered the line plot of wave intensity variation. Please find the panel h of the revised Fig. 1.